# GeoReward: Mitigating Contextual Variable Overestimation in Vision-Language Models for Cross-Market Preference Prediction

Shuo Liu [1]  Huixiang Cai [1]  Weiru Zhang [1]  Xiaoyi Zeng [1]

## Abstract

Vision-language models (VLMs) excel in many multimodal tasks but remain prone to a subtle yet impactful failure mode: they tend to overestimate dominant visual-textual cues while underestimating sparse but decision-critical contextual variables. This issue, which we term Contextual Variable Overestimation (CVE), becomes particularly evident in real-world applications such as predicting advertisement image preferences across diverse geographic markets. For instance, when a VLM (e.g., Qwen2-VL) is asked to choose between two product images tailored for different countries (e.g., Korea vs. France), it often defaults to a consistent output (e.g., always selects "A"), ignoring ground-truth regional variations. This collapse occurs because pervasive high-volume signals, such as product attributes and dense image patches, overwhelm the few but critical tokens that encode market-specific context (e.g., country names). To address CVE, we first collect a new multimodal dataset of real advertising creatives and their click-through performance across multiple countries. We then introduce GeoReward, a reward model designed to predict ad image preferences across diverse geographic markets. GeoReward integrates three purpose-built mechanisms: (1) Market-Aware Retrieval Augmentation, which retrieves and injects region-aligned preference signals during training to sharpen localization awareness. (2) Context-Guided Visual Modulation, a lightweight adapter that dynamically adjusts visual representations using textual country embeddings, enabling fine-grained regional adaptation. (3) Selective Sensitivity Loss, an objective that applies heightened penalties for context-specific mispredictions,

sharpening the model's focus on critical variables. Furthermore, we demonstrate how GeoReward can guide the fine-tuning of RL for a VLM to generate background designs for text-to-image models (e.g., SDXL), producing market-aware advertising creatives. Experiments validate that our framework mitigates CVE and outperforms existing baselines. This work not only diagnoses a systematic bias in VLMs toward dominant perceptual features but also delivers a targeted solution for applications where sparse contextual variables govern decision-making. Code is available at https://github.com/liushuo -hue/GeoReward.git.

## 1. Introduction

Vision-language models (VLMs) (Wang et al., 2024; Bai et al., 2025; Zeng et al., 2024) have emerged as a cornerstone of modern artificial intelligence, demonstrating remarkable proficiency across a broad spectrum of multimodal tasks, including visual question answering, image captioning, and complex reasoning about visual scenes. Their ability to learn powerful multimodal representations has been used as multimodal reward models (RMs) (He et al., 2024; Liu et al., 2025a; Xu et al., 2026; Zang et al., 2025; Xiong et al., 2025), which can provide crucial reward signals to guide model training (Ouyang et al., 2022; Rafailov et al., 2023; Schulman et al., 2017; Wang et al., 2024; Liu et al., 2025a) and inference (Gulcehre et al., 2023; Snell et al., 2025). However at the heart of deploying robust Vision-Language Models (VLMs) in real-world, instruction-sensitive scenarios lies a fundamental and under-addressed problem: Contextual Variable Overestimation (CVE). This phenomenon is not merely a performance bug but a systemic flaw in how VLMs integrate multimodal information. When processing an input stream, the model's autoregressive probability chain, $P(\mathbf{x}) = \prod_t P(x_t|x_{<t})$, becomes dominated by high-volume, frequently occurring variables (e.g., dense image patches, verbose product descriptions). Consequently, sparse yet critical variables (e.g., a single country name specifying the target market) are statistically drowned out during attention-based fusion. This leads to a catastrophic collapse

[1]Alibaba International Digital Commerce Group, Hangzhou, China. Correspondence to: Weiru Zhang <weiru.zwr@alibaba-inc.com>.

*Proceedings of the $43^{rd}$ International Conference on Machine Learning*, Seoul, South Korea. PMLR 306, 2026. Copyright 2026 by the author(s).

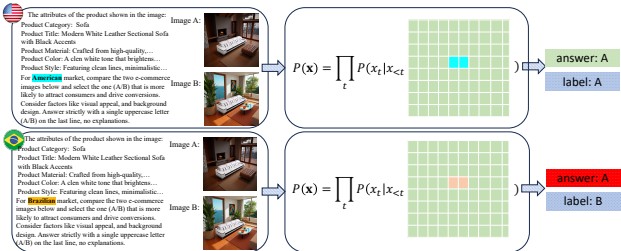

*Figure 1.* Contextual Variable Overestimation (CVE): The only differing variable (blue or orange squares) shown in the two examples is the country names, which are sparse but critical. Other high-volume variables (green squares shown in the image) are dominated, including the same product attributes and the same two images. The model's autoregressive decision-making process collapses. It produces a market-invariant prediction.

in task-specific sensitivity, rendering the model invariant to the very instructions it should follow, as exemplified by its failure in cross-market preference prediction. Specifically, as shown in Figure 1, the only differing variable (blue or orange squares) shown in the two examples is the country names, which are sparse but critical. Other high-volume variables (green squares shown in the image) are dominated. Models such as Qwen-VL (Bai et al., 2023) are required to discern nuanced preferences between two images (A and B) depicting the same product across distinct markets, for instance, comparing consumer choices between the American market and the Brazilian market. Despite clear empirical evidence that human preferences vary significantly by region, VLMs frequently allocate sufficient sensitivity to the critical variable and exhibit a collapse in their decision-making process, defaulting to a single output choice (e.g., persistently selecting "A") irrespective of the target market.

To study this, we first collect a dataset, called Multi-Country Ad Click Preference (MACP), a real-world dataset of advertising image click-through preferences across multi-country markets. Our dataset contains 823K training samples and 180K test samples involving 10 countries. The number of samples from different countries is balanced. Each sample includes two different images of the same product, their Click-Through Rate (CTR), and the detailed product information, including titles and other relevant attributes. These samples are collected from the same e-commerce platform, ensuring consistency in the data source and characteristics.

We then reveal that effectively mitigating CVE necessitates a paradigm of isolated and controllable fusion of multi-granularity knowledge flows. The key lies in a system that simultaneously achieves three objectives: (1) Precise injection of external dynamic knowledge to introduce evidence directly relevant to the sparse critical variable; (2) Interference-proof modulation of internal solidified knowledge to protect foundational capabilities from degradation by high-volume variables when adapting to new contexts;

(3) Differentiated, root-cause learning signal guidance to instruct the model in reconstructing the decision weight of the sparse critical variable at the representation level. We formalize this within a Bayesian framework, showing the optimal approximation is a hierarchical evidence integration model with adaptive gating (see Appendix A).

This insight leads to our proposed GeoReward, designed to predict ad image preferences across diverse geographic markets. It is an architectural instantiation of the above principle. GeoReward implements a triple-gated pipeline: a Retrieval Gate (Market-Aware Retrieval-Augmented Generation) filters external knowledge for relevance and sparsity, a Consolidation Gate (Context-Guided Visual Modulation) dynamically adjusts visual representations using textual country embeddings, enabling fine-grained adaptation to diverse market-specific characteristics, and a Sensitivity Gate (Selective Sensitivity Loss) adaptively applies varying penalties based on the focus of features such as country, image, and product when a prediction error occurs.

In cross-country ad preference prediction, GeoReward significantly alleviates CVE, outperforming any baseline using single or dual components, and improving prediction accuracy by over $15.75\%$ on the Multi-Country Ad Click Preference (MACP) dataset while maintaining robustness across 10 diverse markets. More importantly, when used as a reward model to fine-tune VLMs via Reinforcement Learning, the optimized model generates highly market-tailored background designs, subsequently guiding text-to-image models (e.g., SDXL (Podell et al., 2024)) to produce product images that genuinely resonate with regional preferences. This demonstrates the universal value and powerful efficacy of our proposed paradigm in solving CVE and enabling reliable contextualized decision-making.

Our contribution is fourfold: First, we identify and formally characterize the pervasive yet understudied problem of Contextual Variable Overestimation (CVE) in VLMs, which leads to catastrophic insensitivity to sparse critical variables in instruction-following scenarios. Second, to rigorously study CVE, we collect and release the Multi-Country Ad Click Preference (MACP) dataset, a large-scale real-world benchmark for cross-market preference prediction. Third, we propose GeoReward, a novel framework that features a triple-gated pipeline for precise external knowledge injection, interference-proof internal knowledge modulation, and root-cause-sensitive guidance, effectively mitigating CVE. Finally, we employ GeoReward as a reward model to fine-tune VLMs via Reinforcement Learning (RL), enabling the generation of market-adapted background designs and assisting text-to-image models in producing images tailored to specific country markets.

**Conflict of Interest Disclosure:** All authors are employed by Alibaba International Digital Commerce Group, which

is affiliated with Alibaba Group (the developer of Qwen-VL2 (Wang et al., 2024), a model evaluated in this paper) and operates the cross-border e-commerce platform from which the Multi-Country Ad Click Preference (MACP) dataset was collected; the proposed GeoReward method will be deployed to optimize advertising on this platform.

## 2. Related Work

**Multimodal Reward Models.** Multimodal reward models play an increasingly critical role in aligning vision, understanding, and generation systems with human preferences. A widely adopted strategy involves fine-tuning vision-language models (VLMs) (Li et al., 2024; Bai et al., 2022), capitalizing on their strong multimodal alignment capacities to acquire reward functions reflective of human judgments. Previous research has investigated reward modeling in the context of visual generation (Liu et al., 2025a; Xu et al., 2026; He et al., 2024; Wang et al., 2025a) and visual understanding tasks (Zang et al., 2025; Xiong et al., 2025). For example, (Ziegler et al., 2020) devises an efficient pipeline for building multimodal preference datasets and utilizes existing high-quality data to train IXC-2.5-Reward, a model capable of accurately assessing outputs from visual understanding tasks. Similarly, (Wang et al., 2025a) gathers human feedback to create a dataset of human-rated videos used to train LiFTCritic, a reward model designed to evaluate how closely generated videos match human expectations. (Wang et al., 2025b) proposes UnifiedReward, a unified reward model that can evaluate image and video generation as well as understanding tasks, showing that collaborative learning across various visual domains leads to significant synergistic improvements. (Wang et al., 2026) presents UnifiedReward-THINK, a unified multimodal reward model based on lengthy chain-of-thought (CoT) reasoning, which facilitates multi-dimensional long-chain reasoning for visual understanding and generation tasks. Despite their promising performance, existing reward models have not considered the setting where instructions contain sparse but critical variables, which frequently lead to imprecise or untrustworthy reward signals. To this end, we propose GeoReward, a reward model that adaptively weights diverse semantic cues (e.g., country names, product attributes, image features), prioritizing sparse yet critical signals that are overshadowed by dominant input features.

**Learning from Human Feedback.** Reinforcement Learning from Human Feedback (RLHF) (Bai et al., 2022; Luong et al., 2024; Ziegler et al., 2020; Ouyang et al., 2022; Jiao et al., 2025; Zhang et al., 2025; Ying et al., 2024; Yang et al., 2024; OpenAI et al., 2024; Shao et al., 2024; Hui et al., 2024) collects human feedback regarding model outputs. The feedback is then used to optimize the generative model via reinforcement learning methods such as

PPO (Schulman et al., 2017), DPO (Rafailov et al., 2023), and GRPO (DeepSeek-AI et al., 2025). The RL applications for VLMs include visual quality assessment (Li et al., 2025), visual perception and reasoning (Liu et al., 2025b), mitigating hallucinations (Sun et al., 2024; Yu et al., 2024), and aligning models with human preferences (Yu et al., 2025; Zhou et al., 2024). To bridge VLMs and T2I models, a classifier (Wu et al., 2023) is trained on human-curated image choices and outputs a human preference score used to adapt the T2I model. Parrot (Lee et al., 2024) jointly optimizes the prompt expansion and T2I model network together via a multi-reward RL approach for improving image quality. CAIG (Chen et al., 2025) first explores the utilization of VLMs for generating advertising images by optimizing for CTR as the objective. Through RL, the CTR reward model is used to finetune VLMs. The finetuned VLMs can generate background designs, which are fed into T2I models to generate an image better aligned with user preferences. However, they noted that a limitation is that their reward model overlooks the preferences of niche market segments, and this lack of personalization can lead to suboptimal experiences for diverse user groups. Moreover, our work can better integrate user preferences across different country markets. We use our GeoReward, trained to overcome CVE, as a preference reward model for fine-tuning a VLM. This allows the generative model to produce background designs optimized for specific country markets that cater to the needs and behaviors of global users.

## 3. Method

### 3.1. Problem Formulation

We formalize the CVE problem in geographically-aware visual preference prediction. Given an input tuple $(I_A, I_B, T, c)$, where $I_A$ and $I_B$ are candidate product images, $T$ is the associated textual description (e.g., product title), and $c$ denotes the target country, the objective is to predict a binary preference label $y \in \{A, B\}$.

### 3.2. Overall Framework of GeoReward

As illustrated in Figure 2, GeoReward is built upon the Qwen2-VL framework and is designed as an instantiation of the gated fusion principle. The framework operates through three synergistic stages: (I) The **Retrieval Gate** performs Market-Aware Retrieval-Augmented Generation (MA-RAG) to inject filtered, country-specific experiential knowledge, producing an augmented prior $\hat{y}_{\text{aug}}$. (II) The **Consolidation Gate** employs a Context-Guided Visual Modulation (CGVM) to adaptively adjust the visual feature stream using country embeddings, achieving interference-proof internal adaptation. (III) The **Sensitivity Gate** utilizes a Selective Sensitivity Loss (SSL) during training to calibrate the model's attention, applying penalties focused on

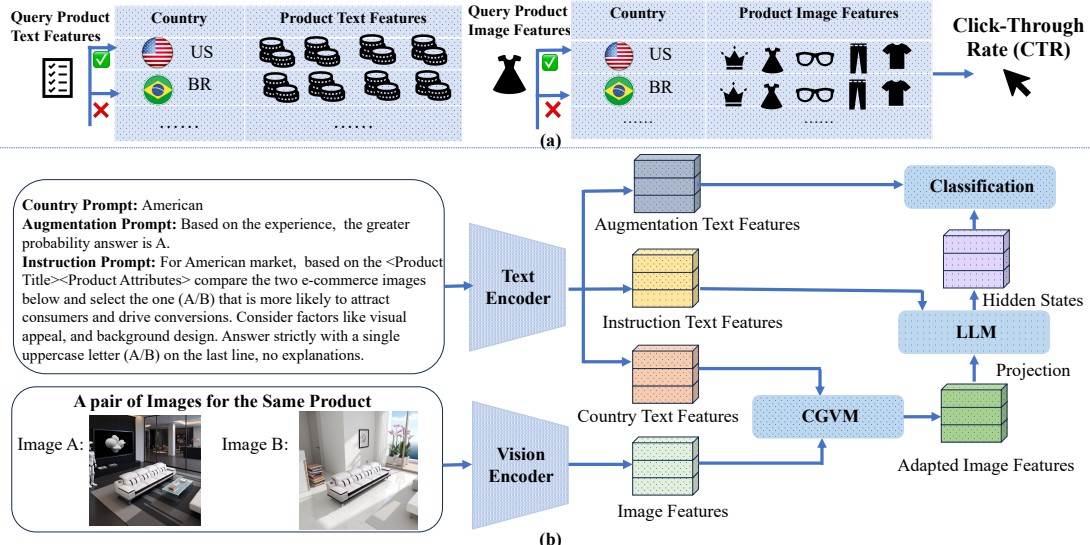

*Figure 2.* GeoReward: Figure (a) illustrates that based on the American knowledge base, the query product text features first retrieve items which have similar product text features, and then the query product image features retrieve the CTR values among the items retrieved in the first stage. Figure (b) depicts the training framework of GeoReward.

under-attended critical features (country, product, image) when predictions are wrong. The final prediction combines the consolidated multimodal representation with the augmented prior. This pipeline ensures that knowledge from different sources and granularities is isolated when necessary and fused in a controlled manner to resolve CVE.

The three stages of GeoReward form a cohesive system that systematically addresses CVE. The **Retrieval Gate** (Stage I) injects sparse, high-level market evidence ($e_{\text{aug}}$) directly into the classifier, providing an explicit prior for $c$. The **Consolidation Gate** (Stage II) operates at the feature level, adaptively tuning the visual stream to align with $c$, thereby preventing the product variable from dominating the representation. Finally, the **Sensitivity Gate** (Stage III) operates at the gradient level, providing a meta-supervision signal that corrects the model's internal attention allocation when it fails, ensuring that the lessons from Stages I and II are effectively absorbed.

### 3.2.1. EXTERNAL KNOWLEDGE INJECTION VIA RETRIEVAL GATE

To address the sparsity of country-specific signals, we design a Retrieval Gate that fetches and aggregates relevant experiential knowledge from the training corpus. This stage, termed Market-Aware Retrieval-Augmented Generation (MA-RAG), operates *before* model training to generate an auxiliary signal $\hat{y}_{\text{aug}}$, which serves as a sparse-variable-informed prior for the main model.

The gate's function is to *filter*: it strictly controls the retrieval scope to prevent label leakage and ensures the retrieved

neighbors are semantically relevant yet distinct from the query. The index is built solely from the training set. For a target country $c$, we construct text and image embedding matrices $\mathbf{T}_c, \mathbf{I}_c$ from the training data. The detailed process is as follows:

**Index Construction:** The retrieval index is constructed solely from the training set. For each country $c \in \{\text{US}, \text{FR}, \text{KR}, ...\}$, we construct a text embedding matrix $\mathbf{T}_c \in \mathbb{R}^{N \times d}$ and an image embedding matrix $\mathbf{I}_c \in \mathbb{R}^{N \times d}$ from the training set, where $N$ is the number of training items.

**Text-based Retrieval** Given the query text embedding $\mathbf{q}_t \in \mathbb{R}^d$, we compute its similarity with all training text embeddings. We then apply a maximum similarity threshold $\tau = 0.7$ (the query text is implicitly excluded because its self-similarity would typically exceed the threshold). We have verified that all the products retrieved by the query text are completely different from the query itself. Finally, we select the top-k highest similarity indices from $\mathcal{C}_t$ to form the text-retrieved neighbor set:

$$\mathbf{s}_t = \mathbf{q}_t \cdot \mathbf{T}_c^T \in \mathbb{R}^N, \mathcal{C}_t = \{j \mid \mathbf{s}_t[j] < \tau\} \quad (1)$$

Finally, we select the top-k highest similarity indices from $\mathcal{C}_t$ to form the text-retrieved neighbor set:

$$\mathcal{N}_t = \text{Top-}k\left(\{\mathbf{s}_t[j] \mid j \in \mathcal{C}_t\}\right) \quad (2)$$

**Image-based Retrieval** Within the subset of image embeddings $\mathbf{I}_{\mathcal{N}_t}$ corresponding to $\mathcal{N}_t$ (i.e., the images of those

training items), we perform retrieval for each candidate image embedding $\mathbf{q}_i^A$ and $\mathbf{q}_i^B$. We apply the same threshold $\tau = 0.7$ to filter out high-similarity items (the query image is implicitly excluded because its self-similarity would typically exceed the threshold). We have verified that all the product images retrieved by the query image are completely different from the query itself. We then select the top-m highest similarity indices. Indices for candidate A are computed as follows:

$$\mathcal{N}_i^A = \text{Top-}m\left(\{\mathbf{s}_i^A[j] \mid j \in \mathcal{C}_i^A\}\right). \tag{3}$$

where $\mathbf{s}_i^A = \mathbf{q}_i^A \cdot \mathbf{I}_{\mathcal{N}_t}^T$, and $\mathcal{C}_i^A = \{j \in \mathcal{N}_t \mid \mathbf{s}_i^A[j] < \tau$.

Similarly, for candidate B:

$$\mathcal{N}_i^B = \text{Top-}m\left(\{\mathbf{s}_i^B[j] \mid j \in \mathcal{C}_i^B\}\right). \tag{4}$$

where $\mathbf{s}_i^B = \mathbf{q}_i^B \cdot \mathbf{I}_{\mathcal{N}_t}^T$, and $\mathcal{C}_i^B = \{j \in \mathcal{N}_t \mid \mathbf{s}_i^B[j] < \tau$.

**Preference Aggregation:** A position-aware weighting scheme is used, with weight $w_i = m - i$ for the i-th retrieved neighbor. Rather than relying on a single best match, this method maintains robustness by aggregating multiple neighbors. The preference scores are computed as:

$$
\begin{aligned}
S_A &= \frac{\sum_{i=1}^m w_i \cdot \mathbb{I}\left[\text{CTR}\left(\mathbf{I}_{\mathcal{N}_i^A}\right) \geq \text{CTR}\left(\mathbf{I}_{\mathcal{N}_i^B}\right)\right]}{\sum_{i=1}^m w_i}, \\
S_B &= \frac{\sum_{i=1}^m w_i \cdot \mathbb{I}\left[\text{CTR}\left(\mathbf{I}_{\mathcal{N}_i^B}\right) > \text{CTR}\left(\mathbf{I}_{\mathcal{N}_i^A}\right)\right]}{\sum_{i=1}^m w_i}.
\end{aligned} \tag{5}
$$

where $\mathbb{I}[\cdot]$ is the indicator function and $\text{CTR}(\cdot)$ denotes the click-through rate knowledge. The final augmented prediction is:

$$
\hat{y}_{\text{aug}} = \begin{cases} A & \text{if } S_A > S_B, \\ B & \text{otherwise.} \end{cases} \tag{6}
$$

This $\hat{y}_{\text{aug}}$ encapsulates gated, market-aware evidence which is subsequently embedded ($\mathbf{e}_{\text{aug}}$) and provided to the model.

### 3.2.2. INTERNAL KNOWLEDGE ADAPTATION VIA CONSOLIDATION GATE

To enable fine-grained adaptation to country contexts without distorting the VLM's pre-trained knowledge, we introduce a Consolidation Gate within the model architecture. This gate, implemented as a Context-Guided Visual Modulation (CGVM) mechanism, dynamically *modulates* the visual feature stream conditioned on the country context.

The gate's function is to *consolidate*: it uses the country embedding to generate a lightweight, adaptive affine transformation for the visual features. This ensures adaptation is context-specific.

Let $\mathbf{c}_i \in \mathbb{R}^d$ denote the mean-pooled embedding vector of the tokenized country name for the $i$-th sample in a batch,

where $d$ is the hidden dimension size. The adaptation parameters are generated by a small feed-forward network, the Context-Guided Visual Modulation:

$$\gamma_i, \beta_i = \text{Split}(\text{CGVM}(\mathbf{c}_i)). \tag{7}$$

where CGVM: $\mathbb{R}^d \to \mathbb{R}^{2d}$ is implemented as:

$$\text{CGVM}(\mathbf{c}_i) = \mathbf{W}_2(\text{ReLU}(\mathbf{W}_1 \mathbf{c}_i + \mathbf{b}_1)) + \mathbf{b}_2. \tag{8}$$

Here, $\mathbf{W}_1 \in \mathbb{R}^{d/2 \times d}$, $\mathbf{b}_1 \in \mathbb{R}^{d/2}$, $\mathbf{W}_2 \in \mathbb{R}^{2d \times d/2}$, $\mathbf{b}_2 \in \mathbb{R}^{2d}$ are learnable parameters. The output is split into two vectors $\gamma_i \in \mathbb{R}^d$ (scale) and $\beta_i \in \mathbb{R}^d$ (shift).

Let $\mathbf{V}_i \in \mathbb{R}^{N \times d}$ represent the sequence of visual features (e.g., N image patch embeddings) corresponding to the i-th sample before integration into the language model's input embedding space. The adapted visual features $\tilde{\mathbf{V}}_i$ are computed via an element-wise affine transformation:

$$\tilde{\mathbf{V}}_i = \gamma_i \odot \mathbf{V}_i + \beta_i. \tag{9}$$

where $\odot$ denotes the Hadamard (element-wise) product. This transformation is applied to the entire set of visual features $\mathbf{V}_i$ associated with the specific country embedding $\mathbf{c}_i$. The modulated visual features $\tilde{\mathbf{V}}_i$ and the instruction text are then fed into the VLM backbone to obtain the contextual hidden states $\mathbf{H}$.

### 3.2.3. LEARNING PROCESS CALIBRATION VIA SENSITIVITY GATE

To explicitly guide the model to reconstruct the importance of the sparse critical variable and other key features, we implement a Sensitivity Gate at the loss level. This gate, realized as a Selective Sensitivity Loss (SSL), *calibrates* the learning signal by applying a penalty only when the model errs *and* neglects specific features.

The gate's function is to *calibrate*: it is conditionally activated by prediction errors ($\mathbb{I}_i = 1$ if prediction is wrong). Upon activation, it computes a penalty based on the relative focus the model allocated to critical tokens—country ($t_c$), product ($t_p$), and image ($t_i$) in the final hidden states $\mathbf{H}$.

Let $\mathbf{H} \in \mathbb{R}^{T \times d}$ denote the hidden states of the final transformer layer, where T is the sequence length and d is the hidden dimension. The hidden states $\mathbf{H}$ are obtained by feeding the adapted visual features $\tilde{\mathbf{V}}$ (stated in 3.2.2) and instruction text features $\tilde{\mathbf{T}}$ into the VLM. Next, for each sample in a batch, we identify the token positions of key input components: the country token $t_c$, product token $t_p$, and image token $t_i$. The focus intensity toward each component is approximated using the L2-norm of their corresponding hidden states, and the penalty terms for country, product,

and image are shown as follows, where $|\cdot|_2$ is the L2-norm:

$$\text{Focus}_c = \frac{\|\mathbf{H}[t_c]\|_2}{\sum_{j=1}^{T}\|\mathbf{H}[j]\|_2}, \quad \mathcal{P}_c = 1 - \text{Focus}_c,$$

$$\text{Focus}_p = \frac{\|\mathbf{H}[t_p]\|_2}{\sum_{j=1}^{T}\|\mathbf{H}[j]\|_2}, \quad \mathcal{P}_p = 1 - \text{Focus}_p, \quad (10)$$

$$\text{Focus}_i = \frac{\|\mathbf{H}[t_i]\|_2}{\sum_{j=1}^{T}\|\mathbf{H}[j]\|_2}, \quad \mathcal{P}_i = 1 - \text{Focus}_i.$$

These penalties are activated only when the model makes an incorrect prediction. For a batch of size $B$, let $\hat{y}_i$ and $y_i$ be the predicted probability and ground-truth label for the i-th sample, respectively. The indicator function for incorrect prediction is:

$$\mathbb{I}_i = \begin{cases} 1 & \text{if } (\hat{y}_i \geq 0.5) \neq (y_i = 1), \\ 0 & \text{otherwise.} \end{cases} \quad (11)$$

The total penalty loss for the batch is computed as:

$$\mathcal{L}_{\text{penalty}} = \frac{1}{B}\sum_{i=1}^{B}\mathbb{I}_i \cdot \left(\mathcal{P}_c^{(i)} + \mathcal{P}_p^{(i)} + \mathcal{P}_i^{(i)}\right). \quad (12)$$

The overall training objective combines the standard binary cross-entropy loss $\mathcal{L}_{\text{BCE}}$ with the gated sensitivity penalty:

$$\mathcal{L} = \mathcal{L}_{\text{BCE}}(\sigma(\mathbf{W}_{\text{classifier}} \cdot (\mathbf{e}_{\text{aug}} + \mathbf{h}_{\text{last}})), y) + \lambda \mathcal{L}_{\text{penalty}}. \quad (13)$$

Following common practice in sequence classification with LLMs (Touvron et al., 2023), we use $\mathbf{h}_{\text{last}}$, the last token of the hidden state as the discriminative representation, since it summarizes the contextual information of the entire input sequence, $\mathbf{e}_{\text{aug}}$ is the text embedding derived from the augmented choice $\hat{y}_{\text{aug}}$ (stated in 3.2.1), $\mathbf{W}_{\text{classifier}}$ refers to the weight matrix in the classifier component of our model, $\sigma$ is the sigmoid activation function, $\lambda$ is a scaling hyperparameter (set to 0.1). This design encourages the model to strengthen its focus on under-attended components when errors occur, thereby improving feature utilization and country-specific decision-making.

### 3.3. Country-Adapted Background Design Generation

To address the challenge of generating market-adapted visual content for cross-border e-commerce, we propose a reinforcement learning-based framework that leverages a GeoReward model to optimize the generation of country-specific background designs. The framework consists of three stages (shown in Figure 3):

Firstly, a Design Generation Model (DGM), implemented as a finetuned Qwen2-VL (Wang et al., 2024), a Vision-Language Model (VLM) trained on the proposed dataset,

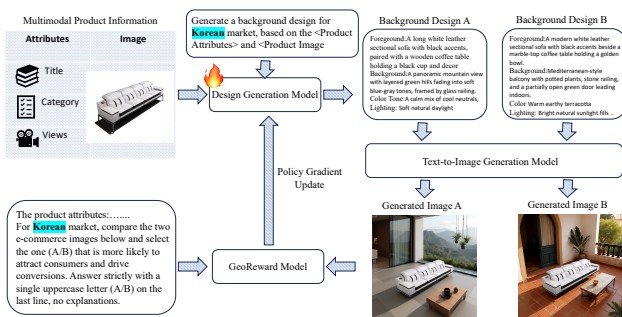

*Figure 3.* Country-Adapted Background Design Generation Framework: Our framework comprises a Design Generation Model that creates country-specific backgrounds, a T2I Generation Model that synthesizes product images, and GeoReward that scores target-country preference and provides optimization feedback to the Design Generation Model.

which generates textual background designs for the target country. The process of training and inference of DGM is:

$$d = \text{DGM}(\text{country}, \text{pro}, \text{I}_{ori}). \quad (14)$$

where country, pro, $\text{I}_{ori}$ represent the target country, the product attributes, and the original product image, respectively. Next, the pair of generated background designs $d_A$ and $d_B$ will be obtained via $d_A = \text{DGM}(\text{country}, \text{pro}, \text{I}_{ori})$ and $d_B = \text{DGM}(\text{country}, \text{pro}, \text{I}_{ori})$.

Secondly, a controlled Text-to-Image (T2I) Generation Model, implemented by integrating a Stable Diffusion Model (Podell et al., 2024) with a ControlNet adapter (Zhang et al., 2023), that allows us to condition the generation process on a control map based on the Canny edge of the product image. The component can enable the generated image to not only align with target market preferences but also adhere to the original product layout. The $\text{T2I}(p, \text{I}_{ori})$ function can be represented as follows:

$$z_{t-1} = \frac{1}{\sqrt{\alpha_t}}(z_t - \frac{1-\alpha_t}{\sqrt{1-\bar{\alpha}_t}}\epsilon_\theta(z_t, t, \tau p, \text{Canny}(\text{I}_{ori}))) + \sigma_t \epsilon. \quad (15)$$

where $z_t$ is the latent representation at timestep $t$, $\text{I}_{ori}$ is the input image, $p$ is the text prompt, $\tau p$ is the text encoder, $\alpha t$, $\bar{\alpha}_t$, $\sigma_t$ are the noise scheduling parameters, $\text{Canny}(\cdot)$ is the canny edge extraction function, and $\text{Decoder}(\cdot)$ can decode the final latent $z_0$ to the generated image $I$ ($I = \text{Decoder}(z_0)$). We will obtain the pair of generated images $I_A$ and $I_B$ via $I_A = \text{T2I}(d_A, \text{I}_{ori})$ and $I_B = \text{T2I}(d_B, \text{I}_{ori})$.

Thirdly, a GeoReward Model (GeoRM) that predicts the country-specific preference for the pair of generated images ($I_A$ and $I_B$). According to the obtained preference choice, the design of a more attractive image is denoted as $d^+$, and the design of a less attractive image is represented as $d^-$. Aiming to finetune the DGM to choose a more attractive

design $d^+$ and reject less attractive ones $d^-$, the feedback signals provided by GeoRM are used to refine the DGM via Direct Preference Optimization (DPO) (Rafailov et al., 2023). Specifically, given an optimization policy model $DGM_\theta$ and a reference model $DGM_{ref}$, the optimization objective is:

$$
\mathcal{L}_{dpo} = -\log \sigma(\beta \log \frac{DGM_\theta(d^+|country, pro, I_{ori})}{DGM_{ref}(d^+|country, pro, I_{ori})}
$$
$$
- \beta \log \frac{DGM_\theta(d^-|country, pro, I_{ori})}{DGM_{ref}(d^-|country, pro, I_{ori})}).
$$
$$(16)$$

where $\sigma$ and $\beta$ are the sigmoid activation function and a regularization parameter, respectively. $DGM_\theta$ and $DGM_{ref}$ are policy and reference models, respectively, where the policy one is optimized while the reference one is frozen. Additionally, the fine-tuned DGM produces product background designs. These designs are then fed into the T2I Generation Model to create product advertising images, ensuring that the generated background designs are tailored to the target country's preferences.

## 4. Experiment

### 4.1. Analysis on GeoReward

**Evaluation Metric.** To evaluate the performance of our GeoReward, we introduce the accuracy and sensitivity metrics. From a business perspective, a model that achieves higher accuracy in predicting which image historically garnered a higher CTR has learned a reward function that directly mirrors past user engagement, and a high sensitivity score ensures that the model's preference judgments are consistent and reliable across country pairs for the same product, which is a prerequisite for trustworthy global deployment. Accuracy measures the proportion of correct predictions, and sensitivity measures the proportion of simultaneous correct predictions across different country combinations, reflecting the market-aware sensitivity of the model's predictions, which is defined as:

$$
\text{Accuracy} = \frac{1}{N} \sum_{i=1}^{N} \mathbb{I}[\hat{y}_i = y_i], \quad \text{Sensitivity} =
$$
$$
\frac{\sum_{i=1}^{M} \sum_{(c_j,c_k) \in \mathcal{C}_2(S_i)} \mathbb{I}[\hat{y}_{i,c_j} = y_{i,c_j} \wedge \hat{y}_{i,c_k} = y_{i,c_k}]}{\sum_{i=1}^{N} |\mathcal{C}_2(S_i)|}.
$$
$$(17)$$

where $N$ represents the total number of samples, $\hat{y}_i$ denotes the predicted class label for the i-th sample, obtained by thresholding the sigmoid normalized logits at $0.5$ and mapping to class labels A, B, $y_i$ corresponds to the ground-truth label, $M$ is represented as total number of unique items, $S_i$ is the set of countries for item $i$, $\mathcal{C}_2(S_i)$ is the set of all 2-combinations of countries in $S_i$, $\hat{y}_{i,c}$ is the predicted answer

for item $i$ in country $c$, $y_{i,c}$ is the ground-truth answer for item $i$ in country $c$, and $\mathbb{I}[\cdot]$ is the indicator function (1 if condition true, 0 otherwise).

The ground-truth labels $y_i$ (A or B) are derived from the CTR data as follows: We have two images (A and B) with their respective CTRs: $CTR_A$ and $CTR_B$. We use a threshold-based approach to account for statistical uncertainty: Compute the relative CTR difference, $\Delta = \frac{|CTR_A - CTR_B|}{\max(CTR_A, CTR_B)}$. We only include pairs where $\Delta \geq \theta$ (threshold, set to 0.1) to ensure meaningful preference distinctions. To mitigate label ambiguity, all pairs with $\Delta < \theta$ are excluded from both the training and evaluation sets. The label $y_i$ is assigned as A if $CTR_A > CTR_B$, and B otherwise, for pairs passing the threshold. The labels $y_{i,c}$ are obtained in the same way.

**Quantitative Results.** As shown in Table 1, experimental results on our MACP benchmark indicate that when the original Qwen2-VL model is fine-tuned on the MACP dataset using a standard approach, the model exhibits complete prediction collapse. The performance degradation primarily stems from its vulnerability to the CVE effect.

**Ablation Studies and Synergy Validation.** We designed exhaustive ablation experiments, not only removing individual components but also testing alternative combinations of components to demonstrate that the synergistic effect of our proposed method.

Removing MA-RAG leads to a 3.56% drop in overall accuracy and 3.44% drop in sensitivity. Using MA-RAG alone reduces overall accuracy and sensitivity by 4.36% and 3.74%, respectively. These highlight the importance of injecting market-specific preference knowledge to guide the model. We also analyze the correlation between the augmented choice $\hat{y}_{aug}$ and ground-truth labels. The Pearson correlation coefficient was measured at 0.67, indicating a moderate relationship, sufficient to provide auxiliary guidance but not dominant enough to overshadow the country token's influence. Removing CGVM causes a more substantial drop of 2.39% in accuracy and a drop of 1.89%. Only using CGVM causes a drop of 4.45% in accuracy and a drop of 3.16%. These underscore the critical role of dynamically modulating visual features based on country embeddings for adapting to local visual preferences. Removing SSL results in a 3.42% accuracy decrease and a 3.37% sensitivity decrease. Using SSL alone reduces accuracy and sensitivity by 4.33% and 3.72%, respectively. These demonstrate that explicitly penalizing the model for under-attending to critical tokens during errors is an effective regularization strategy.

Removing all modules yields the worst performance (55.60% accuracy, 36.73% sensitivity), whereas the full model achieves the best results (60.37% accuracy, 40.84%

*Table 1.* Comparison of accuracy performance across different reward models on MACP. Higher is better for both accuracy and sensitivity, and their units are percentage (%).

| Model | MA-RAG | CGVM | SSL | Accuracy | Sensitivity | BR | CL | ES | FR | KR | JP | US | MX | AU | SA |
|---|---|---|---|---|---|---|---|---|---|---|---|---|---|---|---|
| Qwen2-VL-7B (finetuned) | ✗ | ✗ | ✗ | 44.61 | 20.82 | 46.96 | 43.71 | 49.76 | 46.15 | 47.78 | 40.99 | 42.99 | 46.66 | 40.92 | 40.16 |
| Qwen2-VL-7B (with FC Head) | ✗ | ✗ | ✗ | 55.60 | 36.73 | 53.69 | 56.57 | 50.86 | 54.41 | 53.07 | 59.10 | 56.94 | 53.75 | 58.47 | 59.21 |
| GeoReward (with only MA-RAG) | ✓ | ✗ | ✗ | 56.01 | 37.10 | 53.88 | 56.93 | 52.01 | 54.78 | 53.79 | 59.16 | 57.49 | 53.82 | 58.96 | 59.22 |
| GeoReward (with only CGVM) | ✗ | ✓ | ✗ | 55.92 | 37.18 | 53.71 | 56.27 | 52.38 | 54.89 | 53.39 | 58.67 | 56.88 | 53.56 | 59.10 | 58.64 |
| GeoReward (with only SSL) | ✗ | ✗ | ✓ | 56.04 | 37.12 | 53.76 | 56.06 | 52.48 | 54.47 | 53.36 | 57.78 | 56.32 | 53.68 | 58.65 | 59.48 |
| GeoReward (w/o MA-RAG) | ✗ | ✓ | ✓ | 56.81 | 37.40 | 54.93 | 57.05 | 53.80 | 55.39 | 54.48 | 59.57 | 57.65 | 55.64 | 59.52 | 60.11 |
| GeoReward (w/o CGVM) | ✓ | ✗ | ✓ | 57.98 | 38.95 | 55.50 | 57.85 | 53.78 | 56.75 | 55.19 | 61.47 | 59.39 | 55.98 | 61.71 | 62.21 |
| GeoReward (w/o SSL) | ✓ | ✓ | ✗ | 56.95 | 37.47 | 54.79 | 57.09 | 52.80 | 56.12 | 54.44 | 61.01 | 58.79 | 55.33 | 58.83 | 60.33 |
| **GeoReward** | ✓ | ✓ | ✓ | **60.37** | **40.84** | **58.70** | **61.12** | **56.33** | **59.38** | **57.88** | **63.54** | **61.82** | **58.72** | **63.30** | **62.93** |

*Table 2.* Comparison of performance across different DGMs on MACP. The unit of GeoReward is percentage (%).

| Model | Metric | Accuracy | BR | CL | ES | FR | KR | JP | US | MX | AU | SA |
|---|---|---|---|---|---|---|---|---|---|---|---|---|
| DGM (w/o RL) | GeoReward | 56.04 | 54.03 | 57.05 | 51.67 | 55.40 | 53.59 | 60.40 | 57.64 | 54.36 | 56.38 | 59.89 |
| DGM | GeoReward | **59.60** | **58.36** | **60.10** | **54.66** | **57.31** | **56.22** | **61.76** | **60.28** | **57.33** | **68.26** | **61.71** |

sensitivity). Its performance is not only higher than any single component or binary combination, but the improvement margin also exceeds the simple arithmetic sum of the independent contributions of each component. This highlights the synergy among MA-RAG, CGVM, and SSL: MA-RAG supplies precise context to resolve knowledge conflicts; CGVM uses this context to efficiently target visual feature transformations; and SSL continually reinforces attention to sparse but critical signals, such as the country variable, during training. All three are indispensable, together constituting the optimal solution for mitigating the CVE problem.

**Performance per Country.** Table 1 shows the accuracy breakdown for each country. GeoReward achieves more balanced and higher performance across all countries compared to baselines. The variances of Qwen2-VL-7B with FC Head and GeoReward are $7.12\%$ and $8.18\%$, respectively. GeoReward's specialized components obtain a more robust adaptation to diverse markets.

### 4.2. Analysis on Country-Adapted Background Design Generation

**Evaluation Metric.** We use GeoReward to evaluate the performance of our Country-Adapted Background Design Generation. We use accuracy as the evaluation metric. It measures the proportion of correct predictions. Specifically, each test sample in the MACP dataset contains a product, its attributes, a target country, and a ground-truth preference image ($I_{gt}$). We use the Design Generation Model (DGM) to produce a predicted background ($d_{pred}$). The design is then fed into the controlled T2I pipeline to generate the corresponding advertising image $I_{pred}$. The fixed,

pre-trained GeoReward model is then invoked to evaluate the pair ($I_{gt}$, $I_{pred}$) conditioned on the same product and country context from the original sample. GeoReward outputs a binary prediction $\hat{y} \in \{gt, pred\}$ indicating its inferred preference. The accuracy is calculated as: Accuracy $= \frac{1}{N} \sum_{i=1}^{N} \mathbb{I}[\hat{y}_i = y_i]$, where $N$ is the number of test samples, $y_i$ denotes the target label pred, and $\mathbb{I}[\cdot]$ is the indicator function. A prediction is counted as correct only when the model's prediction favors the predicted image ($I_{pred}$), implying that the model is considered effective solely when the image it generates surpasses the corresponding ground-truth (human-preferred) image.

**Quantitative Results.** As shown in Table 2, images generated by our method consistently achieve higher GeoReward scores across all evaluated countries, indicating that the optimized DGM produces backgrounds that yield images better aligned with country-specific preferences.

**Case Study.** Figure 4 in the appendix presents a case study for two products for five targeted country markets. This qualitative analysis demonstrates our method's capability to produce highly customized visual content that aligns with the preferences of diverse global markets, and our model can capture nuanced, country-specific visual preferences, validating its effectiveness in mitigating CVE and enabling tailored content generation for global markets.

## 5. Conclusion and Future Work

This work identifies and addresses the Context Variable Under-estimation (CVE) problem in Vision-Language Models (VLMs), where models fail to respond to variables that are sparse in the input space but crucial to the instruction. We propose a novel training framework that effectively mit-

igates CVE by integrating components such as retrieval-augmentation, context-guided visual modulation, and selective sensitivity loss. Evaluation of the newly introduced MACP dataset shows that our resulting GeoReward model achieves significant improvements in cross-national preference prediction accuracy. Furthermore, we demonstrate its practicality by using it as a reward signal to optimize background design generation tailored to specific markets. This research lays the groundwork for enhancing sensitivity to key but sparse variables in multimodal reward models. The CVE problem is unlikely to be limited to the country variable; future work should investigate other sparse instructional signals such as user age, gender, or seasonal context. Developing a unified benchmark that systematically varies the sparsity and criticality of control variables would accelerate progress.

## 6. Discussion and Limitations

(i) **Discussion on Cultural Image Generation.** Recent researcher (Khanuja et al., 2024) has explored methods to edit the image to a target culture. The process of adapting visual content for a target culture is a canonical CVE task. The target culture functions as a sparse critical variable that is easily overwhelmed by the dominant visual features of the source image. Our pipeline, which chains a Design Generation Model (DGM), a T2I generator, and GeoReward, directly aligns with the transcreation workflow: DGM produces culturally adapted background descriptions, the T2I model generates the corresponding images, and GeoReward evaluates the results through a culture-sensitive reward signal. By maximizing this culture-sensitive score, GeoReward guides the generative model toward culturally appropriate outputs. Furthermore, the DPO-based fine-tuning of our DGM provides a ready-made template for similar tasks, demonstrating how preference optimization with a CVE-aware reward model can bridge the gap between generic generation and culturally targeted content.

(ii) **Discussion on Cultural Representativeness Metrics.** Recent research (Rege et al., 2025) has utilized VLM-driven image-text alignment metrics to score the degree of cultural representativeness by measuring alignment with simple country prompts or between hierarchical prompts. GeoReward can also be adapted as a scorer for evaluating the cultural representativeness of generated images. Given a cultural concept, the model compares two images and predicts which one better reflects the target culture. This can be trained on triplets of the form (concept, Image A, Image B, human preference label), where the cultural variable serves as the sparse critical cue. The resulting scorer can then be used to judge T2I models via pairwise image comparisons, offering a principled way to quantify and optimize the cultural attribute depiction ability of generative models.

(iii) **Discussion on Cold-Start Robustness.** While our main experiments assume reasonably balanced data across countries, real-world deployment frequently encounters markets with severely limited historical data. To assess GeoReward's resilience in such a data-scarcity regime, we conducted a simulated cold-start experiment. We progressively reduced the training data for a specific country, Chile (CL), from $100\%$ down to $30\%$ of its original volume, while keeping the training data for all other countries unchanged. Evaluating on Chile's held-out test set, we find that GeoReward retains over $85\%$ of its full-data accuracy ($52.84\%$ with only $30\%$ data, compared to $61.12\%$ with the complete training set). This demonstrates that GeoReward does not strictly depend on abundant per-country samples. In actual deployment, when entering a new market, the platform can further boost initial performance by explicitly drawing on insights from culturally or economically analogous regions, gradually accumulating market-specific data to refine predictions.

(iv) **Limitations.** Despite the promising results, several limitations remain. First, our framework targets a single sparse critical variable (the target country); scaling to multiple simultaneous sparse variables (e.g., age and country) may require more sophisticated attention control or ensemble strategies, which we leave for future work. Second, A comprehensive benchmark spanning diverse domains is still missing, which restricts a full assessment of CVE's generality and our method's adaptability.

## Impact Statement

This research improves the reliability and trustworthiness of vision-language models by mitigating Contextual Variable Overestimation (CVE) and boosting sensitivity to sparse yet critical instructional variables. Ethically, we emphasize the importance of responsible data practices in global AI applications. Societally, our framework can enable more effective and personalized cross-country-market digital marketing, and the underlying methodology offers a pathway to build AI systems that are more robust to subtle contextual shifts in various decision-making domains.

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

# A. Theoretical Derivation: GeoReward Framework for Hierarchical Evidence Integration with Adaptive Gating

Below is a formal mathematical explanation of the proposed Hierarchical Evidence Integration Model with Adaptive Gating, derived from first principles using a Bayesian framework and variational inference. This derivation provides the theoretical foundation for addressing the Contextual Variable Overestimation (CVE) problem.

## A.1. Problem Formalization: The Failure Mode of Standard Models (CVE)

Let the multimodal input be $\mathcal{I} = (V, T)$, where $V$ denotes visual data (e.g., image patches) and T denotes textual data. The text T contains both high-volume contextual variables $\mathbf{c}$ (e.g., product descriptions) and sparse-critical variables $\mathbf{z}$ (e.g., target country). The model must predict a decision y (e.g., image preference).

A standard autoregressive Vision-Language Model (VLM) factorizes the probability as:

$$P(y \mid \mathcal{I}) = \prod_{t=1}^{L} P(y_t \mid y_{<t}, \mathcal{I}). \tag{18}$$

Its core is a feature fusion function $f_\theta$ (e.g., Transformer layers (Vaswani et al., 2017)):

$$\mathbf{h} = f_\theta\big(\operatorname{Enc}_V(V), \operatorname{Enc}_T(T)\big), P(y_t \mid y_{<t}, \mathcal{I}) = \operatorname{Softmax}(\mathbf{W}\mathbf{h}_t). \tag{19}$$

The root of CVE: In the fusion process $f_\theta$, features $\operatorname{Enc}_T(\mathbf{c})$ (high magnitude) and $\operatorname{Enc}_T(\mathbf{z})$ (low magnitude) are aggregated indiscriminately. Due to the dominance of $\operatorname{Enc}_T(\mathbf{c})$, the influence of $\mathbf{z}$ is statistically drowned out in the final representation $\mathbf{h}$. Mathematically, this is equivalent to approximating the true conditional distribution with a degenerate one that ignores $\mathbf{z}$:

$$P(y \mid \mathcal{I}) = P(y \mid \mathbf{c}, \mathbf{z}) \approx P(y \mid \mathbf{c}). \tag{20}$$

The chain-rule dependency on $\mathbf{z}$ is broken, leading to market-invariant predictions.

## A.2. Bayesian Ideal: A Hierarchical Evidence-Integration View

An optimal Bayesian (Bayes & Price, 1763) decision-maker would compute the posterior:

$$P(y \mid V, T) = \frac{P(V, T \mid y)P(y)}{P(V, T)}. \tag{21}$$

To explicitly model the hierarchy between sparse-critical variables and dense contextual data, we introduce a latent variable decomposition. We assume the observed data (V, T) is generated by a high-level intent variable $\mathbf{z}$ (e.g., target market) and low-level implementation variables $\mathbf{x}$ (e.g., visual and textual content features).

Thus, the ideal generative process can be refactored as:

$$P(y \mid V, T) \propto P(y) \sum_{\mathbf{z}} P(\mathbf{z} \mid T) \int_{\mathbf{x}} P(y \mid \mathbf{z}, \mathbf{x}) \, P(\mathbf{x} \mid V, T, \mathbf{z}) \, d\mathbf{x}. \tag{22}$$

This formulation embodies our hierarchical evidence-integration principle:

- $P(\mathbf{z} \mid T)$: Identify and separate the sparse-critical variable $\mathbf{z}$ from the text T. This corresponds to high-level evidence extraction.

- $P(\mathbf{x} \mid V, T, \mathbf{z})$: Re-interpret the low-level evidence $\mathbf{x}$ conditioned on the high-level intent $\mathbf{z}$. This is context-guided modulation of perception.

- $P(y \mid \mathbf{z}, \mathbf{x})$: Make the final decision based on the integrated hierarchical evidence.

The summation and integral are intractable for complex, high-dimensional data. We need a tractable approximation.

## A.3. Variational Approximation and the Emergence of Adaptive Gating

We introduce a variational distribution $Q_\phi(\mathbf{z}, \mathbf{x} \mid V, T)$ to approximate the true posterior $P(\mathbf{z}, \mathbf{x} \mid V, T, y)$. Following variational inference, we maximize the Evidence Lower Bound (ELBO) (Jordan et al., 1999):

$$\log P(y \mid V, T) \geq \mathbb{E}_{Q_\phi}\big[\log P_\theta(y \mid \mathbf{z}, \mathbf{x})\big] - D_{\mathrm{KL}}\big[Q_\phi(\mathbf{z}, \mathbf{x} \mid V, T) \parallel P(\mathbf{z}, \mathbf{x} \mid V, T)\big]. \tag{23}$$

To reflect our hierarchical design, we make a structured factorization assumption:

$$Q_\phi(\mathbf{z}, \mathbf{x} \mid V, T) = Q_{\phi_z}(\mathbf{z} \mid T) \cdot Q_{\phi_x}(\mathbf{x} \mid V, T, \mathbf{z}). \tag{24}$$

This factorization is the mathematical core of our architecture:

- $Q_{\phi_z}(\mathbf{z} \mid T)$: A module that extracts the critical variable $\mathbf{z}$ from text. This can be implemented as a light-weight parser or a retrieval gate.

- $Q_{\phi_x}(\mathbf{x} \mid V, T, \mathbf{z})$: A module that generates the low-level representation conditioned on $\mathbf{z}$. This directly instantiates the context-guided visual modulation.

With this, the first term of the ELBO becomes:

$$\mathbb{E}_{\mathbf{z} \sim Q_{\phi_z}, \mathbf{x} \sim Q_{\phi_x}}\big[\log P_\theta(y \mid \mathbf{z}, \mathbf{x})\big]. \tag{25}$$

How Adaptive Gating Emerges: The practical implementation of $Q_{\phi_x}(\mathbf{x} \mid V, T, \mathbf{z})$ requires a gating mechanism. For instance, it can be a conditional feature transformation:

$$\mathbf{x} = \mathrm{Modulate}\big(\mathrm{Enc}_V(V), \mathrm{Enc}_T(T); \mathbf{g}\big), \mathbf{g} = \sigma\big(\mathbf{W}_g[\mathrm{Enc}_T(T)_\mathbf{z}; \mathrm{Pool}(\mathrm{Enc}_V(V))]\big), \tag{26}$$

where $\mathbf{g}$ is the adaptive gating vector, dynamically computed from the high-level variable encoding $\mathrm{Enc}_T(T)_\mathbf{z}$ and a summary of low-level features. The sigmoid function $\sigma$ ensures the gate values are in [0,1]. This gating vector modulates how low-level features are fused, realizing conditioning $\mathbf{x}$ on $\mathbf{z}$.

## A.4. Final Objective and Selective Sensitivity Loss

Maximizing the ELBO leads to our final training objective, which can be expanded and regularized as:

$$\mathcal{L} = \underbrace{\mathbb{E}_{Q_{\phi_z} Q_{\phi_x}}\big[-\log P_\theta(y \mid \mathbf{z}, \mathbf{x})\big]}_{\text{Prediction Loss}}$$
$$+ \lambda_1 \cdot \underbrace{D_{\mathrm{KL}}\big[Q_{\phi_z}(\mathbf{z} \mid T) \parallel P(\mathbf{z})\big]}_{\text{Sparsity/Prior Constraint}} \tag{27}$$
$$+ \lambda_2 \cdot \underbrace{D_{\mathrm{KL}}\big[Q_{\phi_x}(\mathbf{x} \mid V, T, \mathbf{z}) \parallel P(\mathbf{x})\big]}_{\text{Representation Regularizer}}.$$

The Selective Sensitivity Loss is inherently implemented by this objective. When a prediction error occurs, the gradient of the first term $(-\log P_\theta(y \mid \mathbf{z}, \mathbf{x}))$ flows back through the network:

- It updates the predictor $P_\theta$.

- It flows into the modulation network $Q_{\phi_x}$, adjusting the gating mechanism $\mathbf{g}$.

- Crucially, it also flows into the extractor $Q_{\phi_z}$, reinforcing or correcting the identification of $\mathbf{z}$.

If an error is attributable to neglecting $\mathbf{z}$, the gradient signal will strongly encourage the gate $\mathbf{g}$ to give higher weight to features relevant to $\mathbf{z}$ in future similar inputs. This process is adaptive; the sensitivity to the critical variable is adjusted based on empirical error.

### A.5. Correspondence to the Proposed GeoReward Framework

This mathematical derivation shows that our proposed architecture is not an ad-hoc assembly but a natural instantiation of the variational principle for solving CVE:

- Market-Aware RAG Module $\approx$ Implements the prior $P(\mathbf{z})$ and aids the approximate posterior $Q_{\phi_z}(\mathbf{z} \mid T)$ by retrieving relevant, market-specific evidence.

- Context-Guided Visual Modulation $\equiv$ The physical implementation of the conditional distribution $Q_{\phi_x}(\mathbf{x} \mid V, T, \mathbf{z})$, where the adaptive gating occurs.

- Selective Sensitivity Loss $\equiv$ The combined effect of the prediction loss and the KL divergence terms within the ELBO framework, which differentially penalizes errors based on the inferred importance of $\mathbf{z}$ and drives the adaptive learning of the gates.

The Hierarchical Evidence Integration Model with Adaptive Gating is derived from a Bayesian formulation of the CVE problem. The need for adaptive gating emerges naturally from the requirement to approximate the conditional distribution $Q_{\phi_x}(\mathbf{x} \mid V, T, \mathbf{z})$ within a tractable variational inference scheme, providing a principled theoretical foundation for the GeoReward framework.

## B. Reproducibility Statement

We confirm that the methodology presented in this paper is fully reproducible. To support transparency and facilitate further research, we will publicly release all data and source code used in our experiments upon acceptance of the paper. The code repository includes detailed instructions for environment setup, training, and evaluation to ensure easy replication of our results.

## C. LLM Disclaimer

We acknowledge the use of Large Language Models (LLMs) in the preparation of this manuscript. Specifically, DeepSeek (DeepSeek-AI et al., 2025) was used solely for two purposes: (1) to assist in literature review by summarizing existing research and identifying relevant papers, and (2) to polish the text for improved fluency and readability. All ideation, theoretical development, experimental design, data analysis, and result interpretation were conducted solely by the authors. The authors take full responsibility for the content, accuracy, and originality of the work presented herein.

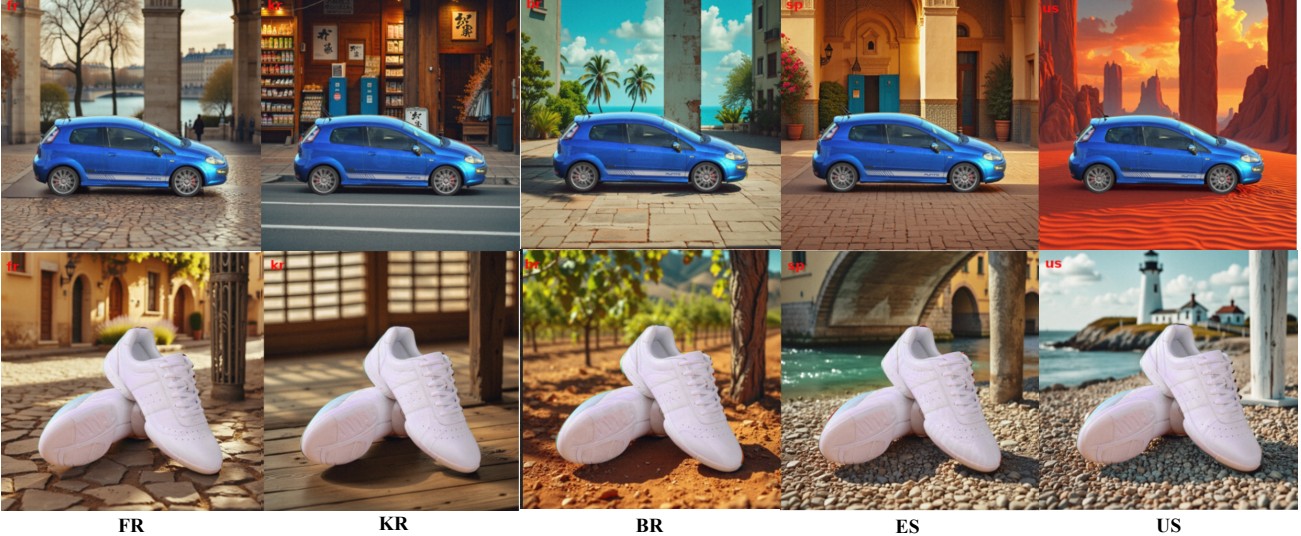

| FR | KR | BR | ES | US |

*Figure 4.* Case Study for two products across five distinct countries.

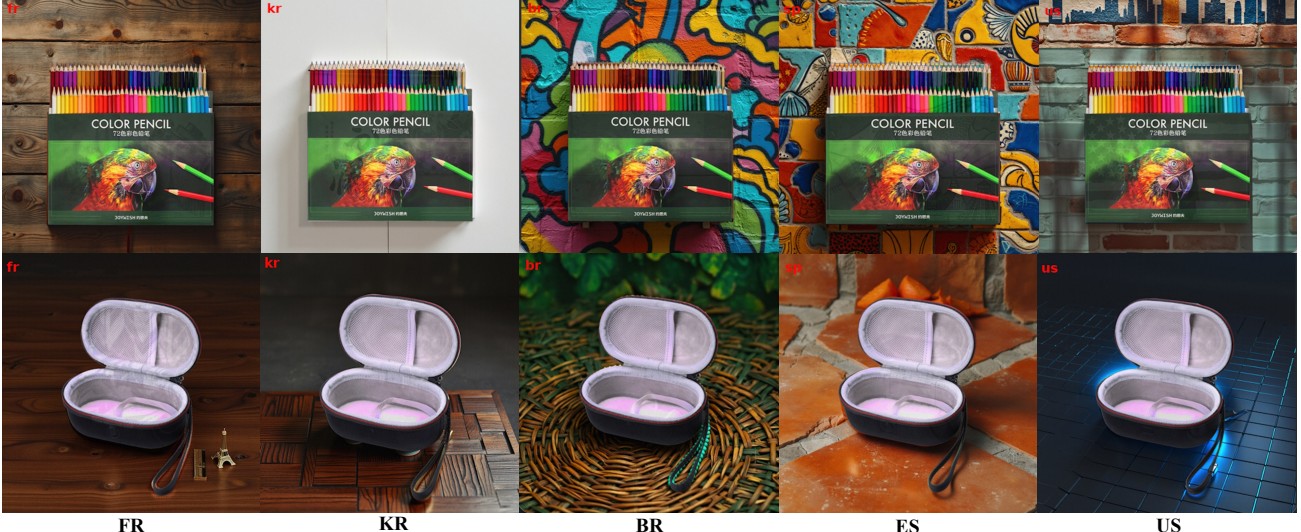

*Figure 5.* Bad Case Study for two products across five distinct countries.

## D. Case Study

This case study investigates product advertising adaptation across multi-country markets by generating location-specific marketing imagery for two products. A compact blue car and a pair of white sneakers are shown in the Figure, across five distinct countries: France (FR), Korea (KR), Brazil (BR), Spain (ES), and the United States (US). This qualitative analysis demonstrates our framework's capability to produce highly customized visual content that aligns with the nuanced aesthetic preferences of diverse global markets, such as Parisian architecture for FR, traditional wooden interiors for KR, tropical coastal vistas for BR, Mediterranean urban textures for ES, and iconic desert or coastal landscapes for US. The study highlights the role of multimodal generative AI in scalable, location-aware marketing design, paving the way for automated, globally distributed visual campaigns that remain sensitive to regional identity and consumer expectations.

Most errors of GeoReward are attributed to extreme visual similarity. When the two candidate images (A and B) are visually nearly identical, the model struggles to discern a preference. As shown in Figure 5, the third and fourth columns in the first row have very similar backgrounds. The images in the first and second columns of the second row also share very similar backgrounds. These findings are valuable for future work, suggesting avenues like incorporating finer-grained visual difference detection.

## E. Determination of the Optimal K-Value in MA-RAG

To determine the optimal k value for top-k retrieval in our MA-RAG system, we employ a decay analysis method based on the cumulative attenuation contribution rate. Specifically, we compute the average cumulative decay of similarity scores across ranking positions from a large-scale retrieval experiment. The k value is set at the point where the cumulative decay contribution rate exceeds a threshold of 80%, indicating that including more results beyond this point yields diminishing returns. This data-driven approach ensures that we capture the majority of relevant information while maintaining efficiency.

## F. Implementation Details

For GeoReward, we employ the Qwen2-VL-7B (Wang et al., 2024) as our foundation model. In the MA-RAG process, the top-k values are 127 and 39 in the text and image retrieval stages, respectively. This training phase takes about 20 hours to complete. All experiments are conducted on a machine equipped with 8 NVIDIA A100 GPUs. To optimize training performance, DeepSpeed and FlashAttention-2 are adopted. We use a per-device batch size of 8, gradient accumulation steps of 2, learning rates of $1e-5, 5e-6, 1e-4, 1e-4$ for the projector, LLM, context-guided visual modulation, and classification head, respectively, a cosine learning rate schedule, and 3 epochs with BF16 mixed-precision enabled. The $\lambda$ is set to 0.1 in SSL. For our T2I generation model, we use Stable Diffusion XL (Podell et al., 2024), enhanced with

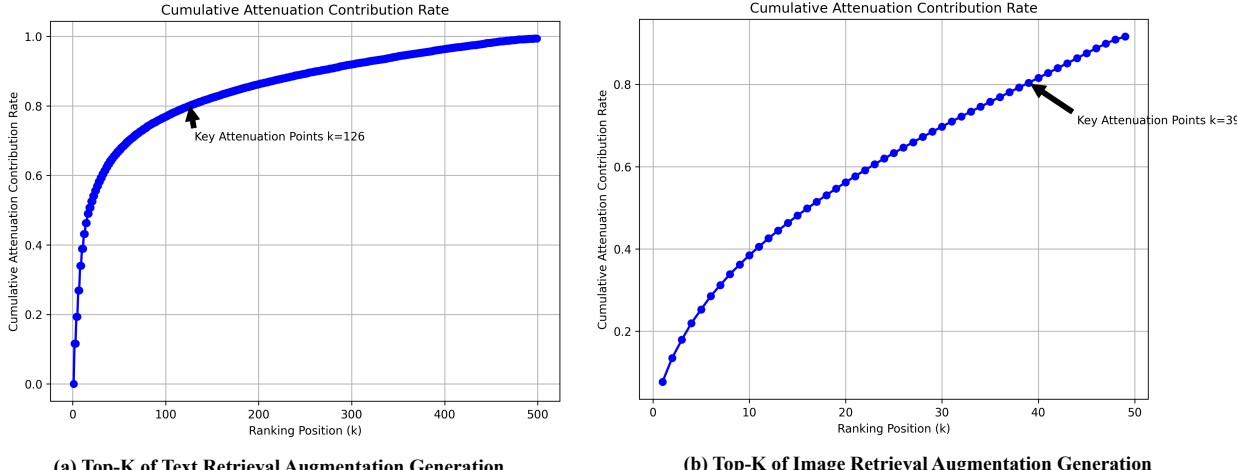

(a) Top-K of Text Retrieval Augmentation Generation       (b) Top-K of Image Retrieval Augmentation Generation

*Figure 6.* The determination of the optimal K-value.

ControlNet (Zhang et al., 2023).

For our RL training, we used Direct Preference Optimization (DPO) to finetune the VLM (Qwen2-VL-7B) using GeoReward as the preference signal. For each product, the DGM generates two background designs $d_A$ and $d_B$. Using the T2I model, we create images $I_A$ and $I_B$, which are scored by GeoReward. The design with the higher reward is treated as the preferred ($d^+$) and the other as rejected ($d^-$). The DPO loss uses a $\beta$ (regularization parameter) value of $0.05$. The policy model ($\text{DGM}_\theta$) was trained for $1$ epoch using the AdamW optimizer with a learning rate of $5e-6$ for the LLM components and $1e-5$ for the multimodal projector. Training used a per-device batch size of $2$ and gradient accumulation over $16$ steps, resulting in an effective batch size of $32$. The max sequence length is $8192$ tokens. LR scheduler is cosine with $10\%$ warmup. Training was conducted on $8$ NVIDIA A100 GPUs using DeepSpeed ZeRO-3. We ran $1$ RL epoch, which involves prompt generation $->$ image generation $\rightarrow$ reward scoring $\rightarrow$ DPO fine-tuning. We will release all training scripts, configuration files, and the final finetuned model weights upon acceptance to ensure full reproducibility.

## G. Augmentation Strategy in MA-RAG

*Table 3.* Comparison of accuracy performance across different augmentation strategies on MACP. Higher is better for both accuracy and sensitivity, and their units are percentage (%).

| Model | Accuracy | Sensitivity | BR | CL | ES | FR | KR | JP | US | MX | AU | SA |
|---|---|---|---|---|---|---|---|---|---|---|---|---|
| Qwen2-VL-7B (with FC Head) | 55.60 | 36.73 | 53.69 | 56.57 | 50.86 | 54.41 | 53.07 | 59.10 | 56.94 | 53.75 | 58.47 | 59.21 |
| Qwen2-VL-7B (with Instruct RAG) | 54.58 | 36.49 | 52.41 | 54.10 | 50.34 | 53.76 | 51.86 | 57.81 | 55.74 | 52.51 | 58.36 | 58.94 |
| Qwen2-VL-7B (with Embedding RAG) | 55.69 | 37.23 | 53.54 | 56.70 | 51.45 | 54.37 | 52.98 | 58.68 | 57.20 | 53.97 | **58.99** | 58.98 |
| Qwen2-VL-7B (with Scaled Embedding RAG) | **56.97** | **38.87** | **53.79** | **57.28** | **52.58** | **55.22** | **54.52** | **60.13** | 57.73 | 54.39 | 56.05 | **59.95** |

Our investigation focuses on the effective incorporation of augmented answers (from Figure 2(a)) into GeoReward. We evaluate two paradigms (Table 3): instruction-based injection ("Qwen2-VL-7B with Instruct RAG") and embedding-based addition ("Qwen2-VL-7B with Embedding RAG") of the answer features to the discriminative features. Since both methods yielded inferior results to the baseline, we subsequently scaled the augmented text features to mitigate potential magnitude mismatches with the discriminative features.

$$\mathbf{E}_{\text{cls}} = \mathbf{h}_{\text{dis}} + \left( \frac{\|\mathbf{h}_{\text{dis}}\|_2}{\|\mathbf{e}_{\text{aug}}\|_2} \right) \cdot \mathbf{e}_{\text{aug}} \tag{28}$$

where $\mathbf{E}_{\text{cls}}$ is used to feed into classification head, $\mathbf{e}_{\text{aug}}$ is the extracted text embedding of augmented answers, and $\mathbf{h}_{\text{dis}}$ is the hidden states of the last token from the VLM.

## H. Real-World Validation through Online A/B Testing

It is crucial to validate the practical impact of our method using real-world user behavior data. To address this and eliminate potential concerns of circular evaluation, we conducted a large-scale online A/B test on a major cross-border e-commerce platform. The test was designed to compare the performance of product images generated using our optimized Design Generation Model (DGM) against those generated by the baseline DGM (without RL fine-tuning). For a consistent product set spanning the 10 countries in our MACP dataset, two advertising image variants were created: one using the background design from our RL-optimized DGM and the other from the baseline DGM. These image pairs were then served randomly to users in their respective target countries. The test ran for one week, accumulating over 6 million impressions in total.

The primary evaluation metric was the Click-Through Rate (CTR), defined as the number of clicks divided by the number of impressions for each image variant. A higher CTR indicates that an image is more effective at attracting user engagement in a real-world scenario. The results, summarized in Table 4, demonstrate consistent and statistically significant improvements in CTR for images generated by our method across all countries. The average CTR improvement was $1.1\%$, with gains ranging from $0.7\%$ to $1.4\%$ depending on the market.

These results provide strong external validation that our GeoReward-optimized framework generates advertising images that are significantly better aligned with country-specific user preferences, leading to higher engagement. This real-world evidence substantiates the practical utility of our approach in mitigating the CVE effect and enhancing cross-market visual content generation.

*Table 4.* CTR improvements of generated images using our optimized DGM vs. the baseline DGM across countries.

| Country | BR | CL | ES | FR | KR | JP | US | MX | AU | SA |
|---|---|---|---|---|---|---|---|---|---|---|
| CTR Improvement (%) | 1.3 | 1.2 | 1.0 | 0.8 | 1.4 | 1.3 | 1.1 | 0.8 | 1.1 | 0.7 |

## I. Discussion on Model Deployment

A crucial aspect of the proposed Market-Aware Retrieval-Augmented Generation (MA-RAG) module is its deployment strategy and associated computational cost, particularly concerning whether it operates online during inference. To clarify, MA-RAG is primarily employed as an offline, pre-computation component. During the training phase, the augmented choice $\hat{y}_{\text{aug}}$ for every sample in the training set is pre-computed offline prior to the commencement of model training. These static, pre-computed augmentations are then integrated into the training process of GeoReward, enriching the model's exposure to market-aligned preferences without incurring retrieval latency during gradient updates.

For the inference or deployment phase, our framework adopts a dual-strategy approach centered on offline-first efficiency. In the standard and recommended deployment setting, MA-RAG is also employed as an offline, pre-computation component. The pre-computed augmentations are then integrated into the inference process of GeoReward. However, the architecture retains the flexibility for online retrieval. When utilizing MA-RAG for online retrieval during inference, processing each sample takes 76 ms, while the offline version delivers faster performance. This makes MA-RAG well-suited for large-scale production environments.

## J. Generalization Experiments

The generalizability of the Contextual Variable Overestimation (CVE) problem is crucial. Below, we address these concerns by: (1) clarifying the fundamental nature of CVE as a general architectural issue in VLMs, (2) presenting new experimental evidence on a different domain and attributes, and (3) explaining how our framework inherently generalizes.

### J.1. The Theoretical Basis for CVE as a General Challenge

The core of the CVE problem is not an artifact of the advertising domain but stems from a fundamental characteristic of autoregressive VLMs. Their probability chain rule, $P(\mathbf{x}) = \prod_t P(x_t|x_{<t})$, coupled with attention-based feature fusion, creates a statistical bias towards high-volume, frequently co-occurring tokens (e.g., detailed product descriptions, hundreds of image-patch tokens). In contrast, instruction-critical variables that are sparse in the input sequence (e.g., "country," "target user age," "target user profession") are easily overwhelmed during this fusion process. The advertising preference task, where "country" is a perfectly sparse yet decisive cue, serves as a clear and controlled manifestation of this underlying

general architectural bias. Consequently, the mitigation principles embedded in our framework, including reinforcing sparse variables with external knowledge (MA-RAG), dynamically modulating representations based on them (CGVM), and penalizing their neglect (SSL), are designed to be general mechanisms applicable beyond the specific case of "country."

## J.2. Experimental Setup on Image Quality Assessment

We leverage the ImageGen-CoT-Reward-5K dataset (Wang et al., 2026) from the UnifiedReward-THINK. This dataset involves evaluating pairs of AI-generated images across multiple quality dimensions (Semantic Consistency, Aesthetics, Authenticity) using a chain-of-thought (CoT) reasoning process. To create a CVE-style evaluation, we systematically reformulate this dataset:

Specifically, we transform the ImageGen-CoT-Reward-5K dataset from the UnifiedReward study into a CVE-style task. In this transformed dataset, the sparse critical variable is the evaluation dimension (e.g., "Semantic Consistency", "Aesthetics", or "Authenticity"), while all other inputs (images, captions, instruction structure) remain identical. The instruction is framed as: "For [Dimension], the caption of the image is [caption]. Given two images, Image A and Image B, which one is better?" This setup mimics the CVE scenario where a sparse textual cue dictates the preference judgment.

The final processed dataset contains $12,811$ training and $5,517$ test samples, spanning the three evaluation dimensions. We compare our adapted GeoReward framework against a strong baseline: a UnifiedReward model finetuned directly on this transformed dataset. This baseline represents a state-of-the-art general-purpose multimodal reward model adapted to the same data.

## J.3. Results and Analysis

As shown in Table 5, our adapted GeoReward model achieves an accuracy of $67.9\%$ and a sensitivity of $24.0\%$, significantly outperforming the finetuned UnifiedReward baseline (accuracy: $62.7\%$, sensitivity: $1.9\%$). This represents an improvement of over $5.2\%$ in absolute accuracy and a $22.1\%$ increase in sensitivity. This demonstrates that the CVE problem is not domain-specific and that our framework generalizes well to other tasks, such as image quality assessment. The key adaptation lies in identifying the sparse critical variable for the new task and configuring the retrieval augmentation (MA-RAG) and feature modulation (CGVM) components accordingly. The Selective Sensitivity Loss (SSL) remains a universally applicable regularization technique.

## J.4. Scalability to Multiple Sparse Variables

Given the potential complexity of instructions that contain multiple sparse critical variables (e.g., "For a [young person] in [France], which image is better?"), Our framework is inherently scalable to such scenarios:

- The Selective Sensitivity Loss (SSL) can naturally be extended to calculate separate penalties for inattention to multiple specified critical tokens (e.g., "young person" and "France"), encouraging the model to attend to all simultaneously.

- For highly complex, multi-factorial decisions, an ensemble approach could be employed. Specialized reward models (e.g., AgeReward, GeoReward), each finetuned to be sensitive to a specific type of sparse variable using our framework, could provide intermediate rewards. These signals could then be aggregated for a final decision via an ensemble strategy.

More effective models for generalizing to multiple sparse variables could be proposed in the future.

We acknowledge that comprehensively benchmarking CVE across numerous domains and variables is a substantial endeavor. To our knowledge, this is the first work to identify and formalize this problem; our primary contribution is to establish its existence and propose a foundational solution framework. We are committed to expanding this line of research and are actively planning to develop a comprehensive benchmark to systematically evaluate CVE across diverse tasks in the future. We will open-source the code and datasets to foster research on this critical issue.

In conclusion, while our current empirical focus is on the "country" variable in advertising, the CVE problem is rooted in a fundamental architectural characteristic of VLMs. Our solution framework provides general, adaptable mechanisms to address it, supported by preliminary cross-attribute validation. We believe this work opens a crucial research direction toward building more robust and context-sensitive multimodal models.

*Table 5.* CVE Generalization to Image Quality Assessment Task (%)

| Model | Sensitivity | Accuracy | Semantic Consistency | Aesthetics | Authenticity |
|---|---|---|---|---|---|
| Baseline | 1.9 | 62.7 | 63.5 | 61.4 | 63.2 |
| Our Model | 24.0 | 67.9 | 65.6 | 67.8 | 70.4 |

## K. Baseline Comparisons

**Domain-Adaptive SFT:** We directly compared GeoReward with supervised fine-tuning (SFT) reward models on our MACP dataset. Table 6 shows that GeoReward (accuracy: $60.4\%$, sensitivity: $40.8\%$) substantially outperforms the Domain-Adaptive SFT baseline (accuracy: $44.6\%$, sensitivity: $20.8\%$). UnifiedReward is essentially a model finetuned on many existing general-purpose reward model datasets, without any changes to the model architecture. To reduce the undesirable interactions between these datasets and MACP, we reproduce UnifiedReward (Wang et al., 2025b) by performing supervised fine-tuning on Qwen2-VL-7B using the MACP dataset. In other words, UnifiedReward is equivalent to Domain-Adaptive SFT in Table 6. This indicates that specialized mechanisms in GeoReward, such as the Market-Aware Retrieval-Augmented Generation, Context-Guided Visual Modulation, and Selective Sensitivity Loss, are crucial for handling CVE, whereas general-purpose models lack such capabilities.

**Prompt Engineering:** We crafted a strong instructional prompt that explicitly rearranges information to foreground the country variable: "For the [Country] market, for the product: [Attributes], compare the two e-commerce images below for the [Country] market and select the one (A/B) that is more likely to attract consumers in the [Country] market?" As shown in Table 6, although higher than the variant without the rearrangement and repetition prompt method (Qwen2-VL-7B (with SFT)), this baseline achieved only $49.77\%$ accuracy (Qwen2-VL-7B (with SFT & Prompt Engineering)), indicating that sophisticated prompting alone cannot resolve the underlying representational imbalance causing CVE.

**FiLM-Style Conditioning Baseline:** This baseline corresponds directly to applying only the Context-Guided Visual Modulation (CGVM) from our framework. It modulates the visual features extracted by the vision encoder using affine transformation parameters generated from textual country embeddings, following the FiLM paradigm (Perez et al., 2017). This tests the sufficiency of dynamic visual feature conditioning based on sparse textual cues. The FiLM-style baseline (GeoReward with only CGVM) achieves an accuracy of $55.92\%$ and a sensitivity of $37.18\%$. While this represents a slight improvement over the base Qwen2-VL-7B model with a standard FC head ($55.60\%$ accuracy), the gain is marginal. This indicates that simply modulating visual features based on country, while beneficial, is insufficient to robustly overcome the overwhelming influence of dominant, high-volume variables in the input sequence.

**Attention-Imbalance Baseline:** This baseline implements a standard attention-balancing regularization. It utilizes only the Selective Sensitivity Loss (SSL) component with the Qwen2-VL-7B backbone. The loss penalizes the model based on the relative attention (approximated by the L2 norm of hidden states) allocated to critical tokens (country, product, image) when a prediction error occurs, aiming to mitigate attention overshadowing. The attention-imbalance baseline (GeoReward with only SSL) reaches a modest improvement to $56.04\%$ accuracy and $37.12\%$ sensitivity. Explicitly penalizing low attention on critical tokens provides a modest boost but fails to fully resolve the prediction collapse across countries.

**Token-Importance Re-Weighting:** We implement a baseline that re-weights the loss only based on country token importance (approximated by gradient norms). While it showed a slight improvement ($55.87\%$ accuracy) over the standard FC Head baseline ($55.60\%$), its performance gain was limited and less stable across countries compared to our integrated approach. This suggests that simply re-weighting the loss is insufficient to address the core representational imbalance in CVE.

**Focal-Loss Discriminator:** This baseline adds a focal loss discriminator (Lin et al., 2017) based on the Qwen2-VL-7B (with FC Head). Our experiments show that while focal loss slightly improves accuracy ($58.10\%$) over the standard FC head ($55.60\%$), it still falls short of GeoReward ($60.37\%$) and exhibits significantly lower sensitivity ($12.02\%$ vs. $40.84\%$). This indicates that simply re-weighting the loss is insufficient to address the core representational imbalance in CVE, as it fails to enhance cross-country consistency.

**Logistic Regression:** This baseline is a classic CTR-prior logistic regression model (LaValley, 2008) with comprehensive feature engineering, including: Text features (TF-IDF vectors from product titles), Categorical features (One-hot encodings for country and category), Interaction features (Explicit country $\times$ category interactions to capture market-specific pref-

erences), and Image features (Differences between image embeddings of candidates A and B). This baseline now serves as a robust non-neural benchmark, directly modeling the problem of CTR prediction. Results show it achieves 52.28% accuracy, significantly below our method (60.37%), confirming that naive feature-based approaches fail to resolve CVE without structured multimodal reasoning.

**Gradient Boosted Decision Tree:** This baseline is a classic CTR-prior GBDT (Friedman, 2001) baseline with country×category interaction features, representing a strong traditional approach for preference prediction. This base-line incorporates: Text features (TF-IDF of item titles), Categorical features (one-hot encoded country and category), Country×category interaction features (both one-hot and statistical features), and Image embedding differences between candidate images. The results demonstrate that our GeoReward model substantially outperforms this strong non-neural baseline (60.37% vs 55.39% accuracy), validating that our approach captures complex multimodal interactions beyond what traditional methods can achieve.

**Country Token Mask:** This baseline is a causal intervention experiment where the country token is explicitly masked (replaced with [MASK]) in the input instruction. This tests the model's dependence on the sparse critical variable (country name). Masking the country token led to a significant accuracy drop (from 60.37% to 57.25%) and a drastic sensitivity reduction (from 40.84% to 12.26%). This demonstrates that GeoReward successfully grounds its predictions in the country variable, and its performance is causally tied to this sparse token. This ablation directly validates CVE as a failure mode and shows that our framework restores sensitivity to the critical variable.

We performed paired t-tests between GeoReward and all baselines. The improvements of GeoReward are statistically significant ($p < 0.01$) in all cases, confirming that our method consistently outperforms alternatives. One of the examples is shown in Section O. These additions reinforce the validity of our conclusions and demonstrate that GeoReward effectively mitigates CVE with high statistical reliability.

*Table 6.* Comparison of accuracy performance across different baselines on MACP. Higher is better for both accuracy and sensitivity, and their units are percentage (%).

| Model | Accuracy | Sensitivity | BR | CL | ES | FR | KR | JP | US | MX | AU | SA |
|---|---|---|---|---|---|---|---|---|---|---|---|---|
| Logistic Regression | 52.28 ± 0.23 | 14.69 | 51.30 ± 0.73 | 52.46 ± 0.73 | 51.17 ± 0.73 | 52.87 ± 0.73 | 51.42 ± 0.73 | 53.99 ± 0.73 | 53.03 ± 0.73 | 52.02 ± 0.73 | 54.35 ± 0.73 | 53.17 ± 0.73 |
| Gradient Boosted Decision Tree | 55.39 ± 0.23 | 15.37 | 54.87 ± 0.73 | 55.39 ± 0.73 | 55.05 ± 0.73 | 55.56 ± 0.72 | 55.24 ± 0.73 | 56.07 ± 0.72 | 55.42 ± 0.73 | 55.05 ± 0.73 | 55.75 ± 0.72 | 55.51 ± 0.72 |
| Qwen2-VL-7B (with Domain-Adaptive SFT) | 44.61 ± 0.23 | 20.82 | 46.96 ± 0.73 | 43.71 ± 0.71 | 49.76 ± 0.72 | 46.15 ± 0.72 | 47.78 ± 0.70 | 40.99 ± 0.71 | 42.99 ± 0.73 | 46.66 ± 0.71 | 40.92 ± 0.73 | 40.16 ± 0.70 |
| Qwen2-VL-7B (with SFT & Prompt Engineering) | 49.77 ± 0.23 | 14.39 | 49.75 ± 0.72 | 48.34 ± 0.72 | 50.43 ± 0.72 | 49.17 ± 0.72 | 49.02 ± 0.72 | 50.01 ± 0.72 | 52.10 ± 0.71 | 49.94 ± 0.72 | 49.74 ± 0.72 | 49.23 ± 0.72 |
| Qwen2-VL-7B (with FC Head) | 55.60 ± 0.23 | 36.73 | 53.69 ± 0.73 | 56.58 ± 0.72 | 50.86 ± 0.73 | 54.41 ± 0.73 | 53.07 ± 0.73 | 59.10 ± 0.72 | 56.94 ± 0.72 | 53.75 ± 0.73 | 58.47 ± 0.72 | 59.21 ± 0.71 |
| Qwen2-VL-7B (with Focal Loss) | 58.10 ± 0.22 | 12.02 | 58.34 ± 0.71 | 59.84 ± 0.70 | 54.97 ± 0.71 | 58.55 ± 0.70 | 58.00 ± 0.71 | 61.88 ± 0.69 | 59.86 ± 0.70 | 56.68 ± 0.71 | 61.58 ± 0.69 | 55.24 ± 0.71 |
| GeoReward (with only CGVM) | 55.92 ± 0.23 | 37.18 | 53.71 ± 0.73 | 56.27 ± 0.72 | 52.38 ± 0.73 | 54.89 ± 0.73 | 53.39 ± 0.73 | 58.67 ± 0.72 | 56.88 ± 0.72 | 53.56 ± 0.73 | 59.10 ± 0.72 | 58.64 ± 0.72 |
| GeoReward (with only SSL) | 56.04 ± 0.23 | 37.12 | 53.76 ± 0.73 | 56.06 ± 0.72 | 52.48 ± 0.73 | 54.47 ± 0.73 | 53.36 ± 0.73 | 57.78 ± 0.72 | 56.32 ± 0.72 | 53.68 ± 0.73 | 58.65 ± 0.72 | 59.48 ± 0.72 |
| GeoReward (with Re-Weighting) | 55.87 ± 0.23 | 36.91 | 58.73 ± 0.72 | 58.93 ± 0.71 | 59.61 ± 0.73 | 53.52 ± 0.72 | 53.95 ± 0.70 | 57.08 ± 0.74 | 51.37 ± 0.73 | 54.28 ± 0.71 | 57.01 ± 0.72 | 54.19 ± 0.73 |
| GeoReward (with Country Token Mask) | 57.95 ± 0.23 | 12.26 | 55.62 ± 0.72 | 59.33 ± 0.72 | 53.63 ± 0.73 | 56.78 ± 0.72 | 55.08 ± 0.73 | 60.75 ± 0.71 | 59.86 ± 0.71 | 56.19 ± 0.72 | 61.22 ± 0.71 | 61.06 ± 0.71 |
| **GeoReward (Ours)** | **60.37 ± 0.23** | **40.84** | **58.70 ± 0.72** | **61.12 ± 0.71** | **56.33 ± 0.72** | **59.38 ± 0.72** | **57.88 ± 0.72** | **63.54 ± 0.70** | **61.82 ± 0.71** | **58.72 ± 0.72** | **63.30 ± 0.70** | **62.93 ± 0.72** |

## L. Mechanism Analysis

To move beyond a phenomenological description and establish a rigorous mechanistic foundation for the Contextual Variable Overestimation (CVE) concept, we conducted a series of analytical experiments. This section delineates how CVE is distinct from related challenges and provides empirical evidence that traces the failure mode from its root cause to our proposed solution through attention analysis, gradient attribution, and causal interventions.

### L.1. Distinction from Related Concepts

**Compared with "Lost-in-the-Middle":** This primarily concerns the positional degradation of information retrieval in long documents. CVE is agnostic to position; even if the country name is placed at the very beginning, its sparse token count relative to the visual and product descriptive tokens leads to its influence being overwhelmed in the fused representation used for the final prediction.

**Compared with "Modality-Volume Imbalance":** While related, prior work often focuses on general alignment or representation fusion between modalities (e.g., text vs. vision). CVE pinpoints a more specific failure mode within a multimodal decision-making task: a specific, low-volume textual cue (the country) is overwhelmed by the combined volume of other textual cues (product attributes) and a high-volume modality (image patches), rendering the model insensitive to a variable that critically determines the ground-truth outcome.

## L.2. Phenomenon: Contextual Variable Overestimation (CVE)

CVE manifests as a systematic failure in VLMs where sparse but critical instructional variables (e.g., country names in multi-country preference prediction) are overwhelmed by dominant high-volume variables (e.g., image patches or product attributes). In our MACP dataset, baseline models like Qwen2-VL exhibit prediction collapse (e.g., consistently outputting "A" regardless of country), resulting in low accuracy and sensitivity. This occurs despite clear ground-truth preference variations across countries, indicating that the model ignores critical sparse cues.

## L.3. Mechanism: Breaking the Chain Rule via Attention and Gradient Analysis

We performed three core analyses comparing a baseline VLM (Qwen2-VL-7B with an FC Head) against our GeoReward model to uncover the mechanistic underpinnings of CVE.

### L.3.1. ATTENTION MAP ANALYSIS REVEALS NEGLECT OF SPARSE CRITICAL TOKENS

We computed the average attention weight assigned to the sparse country token (e.g., "US") in the final transformer layer.

- Baseline VLM: The country token received a negligible mean attention weight of $0.05 \pm 0.01$.
- GeoReward: The attention weight increased significantly to $0.08 \pm 0.03$.

The baseline model fails to allocate sufficient computational focus to the critical sparse variable. The Context-Guided Visual Modulation (CGVM) and Selective Sensitivity Loss (SSL) in GeoReward successfully recalibrate the attention mechanism, directly mitigating the "overwhelm" by strengthening the chain-rule link $P(x_t|x_{<t})$ for the country token.

### L.3.2. GRADIENT ATTRIBUTION ANALYSIS CONFIRMS UNDERLYING INSENSITIVITY

Using Integrated Gradients, we quantified the contribution of the country token to the final prediction output.

- Baseline VLM: The country token contributed merely $6\%$ to the total gradient norm.
- GeoReward: Its contribution rose to $25\%$.

This demonstrates that in standard VLMs, the gradient signal for the sparse critical variable is vanishingly weak during training, preventing effective learning of its importance. Our framework amplifies this signal, ensuring the model's parameters become sensitive to the country variable.

### L.3.3. INTERVENTION EXPERIMENTS ESTABLISH CAUSAL LINK

We performed a causal intervention by ablating (masking) the country token from the input. Table 7 shows the results of intervention experiments.

- Baseline VLM: Ablation caused a negligible performance drop (sensitivity: from $36.73\%$ to $31.48\%$), confirming its predictions were already invariant to the country.
- GeoReward: Ablation led to a substantial decrease (sensitivity: from $40.84\%$ to $12.26\%$).

This experiment establishes a causal link. The country token becomes a decisive factor for GeoReward's decision-making, whereas it is functionally ignored by the baseline. The sharp drop in sensitivity (which measures cross-country consistency) is particularly telling, proving that GeoReward has learned to meaningfully condition its output on this previously overlooked variable.

### L.3.4. CONTROLLED SYNTHETIC EXPERIMENTS ISOLATE THE CVE EFFECT:

To isolate the CVE effect, we created a synthetic variant of our task where we systematically increased the "weight" of the critical variable by repeating the country token in the instruction. The baseline model's performance showed only marginal improvement (accuracy from $44.61\%$ to $49.77\%$) even as the token count (one country token to five country tokens) for the critical variable increased. This confirms that CVE is not merely a problem of token count but a fundamental failure in the model's sensitivity and representational integration of sparse critical information.

## L.4. Solution: GeoReward Framework

Our framework directly counteracts CVE by reinforcing the model's sensitivity to sparse critical variables through three novel components:

**Market-Aware Retrieval-Augmented Generation (MA-RAG):** Injects market-specific preferences into training, augmenting the sparse country signal with similar product knowledge. This explicitly conditions the chain rule on country-aware data, raising accuracy by $3.56\%$ in ablations.

**Context-Guided Visual Modulation (CGVM):** Dynamically modulates visual features using country embeddings, ensuring that image representations are transformed based on sparse textual cues. This restores the chain rule's dependence on country tokens by aligning visual and textual pathways, contributing $2.39\%$ to accuracy gains.

**Selective Sensitivity Loss (SSL):** Penalizes under-attention to critical tokens during errors, directly optimizing attention distributions via a regularization term. This improves sensitivity by $3.37\%$.

## L.5. Conclusion of Mechanism Analysis

These analyses collectively validate the CVE phenomenon at a mechanistic level by spanning attention distribution, gradient flow, causal intervention, and controlled synthesis. Our GeoReward framework directly counteracts these mechanisms, which completes the "phenomenon $\rightarrow$ mechanism $\rightarrow$ solution" loop, providing a solid foundation for the CVE concept and the efficacy of our approach.

*Table 7.* Performance Comparison Before and After Country Token Ablation.

| Model | Accuracy | Sensitivity | BR | CL | ES | FR | KR | JP | US | MX | AU | SA |
|---|---|---|---|---|---|---|---|---|---|---|---|---|
| Baseline (Ablated) | 49.81 | 31.48 | 49.82 | 50.31 | 48.79 | 49.21 | 48.25 | 50.63 | 51.22 | 49.23 | 49.80 | 50.75 |
| Baseline (Full) | 55.60 | 36.73 | 53.69 | 56.57 | 50.86 | 54.41 | 53.07 | 59.10 | 56.94 | 53.75 | 58.47 | 59.21 |
| GeoReward (Ablated) | 57.25 | 12.26 | 55.08 | 58.26 | 53.46 | 56.58 | 54.64 | 59.90 | 58.78 | 55.75 | 59.73 | 60.35 |
| GeoReward (Full) | **60.37** | **40.84** | **58.70** | **61.12** | **56.33** | **59.38** | **57.88** | **63.54** | **61.82** | **58.72** | **63.30** | **62.93** |

# M. Synergistic Design for CVE Mitigation

The core novelty of our work lies not in the invention of entirely new components, but in their principled and synergistic integration to address the newly identified Contextual Variable Overestimation (CVE) problem. The three core modules, Market-Aware Retrieval-Augmented Generation (MA-RAG), Context-Guided Visual Modulation (CGVM), and Selective Sensitivity Loss (SSL), combat CVE at distinct but complementary levels of the model pipeline, forming a holistic solution.

- Data-level amplification (MA-RAG): By retrieving and injecting click-through preferences aligned with the target market, MA-RAG directly amplifies the signal of the critical country variable. It provides an external, market-aware prior that forces the model to prioritize localized relevance, counteracting the dominance of generic high-volume features.

- Feature-level adaptation (CGVM): Operating inside the vision encoder, CGVM dynamically modulates visual representations using learned country embeddings. This ensures that the sparse critical variable actively conditions perceptual processing, preventing visual features from being treated uniformly and enabling fine-grained adaptation to market-specific visual preferences.

- Optimization-level regularization (SSL): During training, SSL penalizes the model when it makes errors while under-attending to the tokens corresponding to critical variables (country, product, image). It acts as a focus-driven regularizer, explicitly encouraging balanced attention allocation and reinforcing the importance of sparse variables in the decision process.

The synergy is closed-loop: as shown in our experiments (Table 8), MA-RAG provides external country-aware signals, CGVM internally aligns visual features with those signals, and SSL ensures robust attention to all critical components during learning. Ablating any module disrupts the loop, producing a measurable drop in performance. For ablation, we train

models that keep only one proposed module at a time (e.g., GeoReward with MA-RAG only, CGVM only, or SSL only). Their performances (56.01%, 55.92%, and 56.04% accuracy, respectively) are superior to the plain finetuned baseline but substantially worse than the full GeoReward (60.37%). This quantitatively demonstrates that each component contributes, but their combined effect is synergistic and necessary for peak performance.

Crucially, the information flow is sequential and reinforcing: The externally retrieved evidence informs the initial bias; the internal modulation refines the feature space to be more country-sensitive, and the selective loss ensures robust and focused learning of these adaptations. This triple-gated hierarchy mirrors a Bayesian evidence integration process, where prior (retrieved knowledge), likelihood (modulated perception), and posterior update (focused learning) are distinctly managed yet collaboratively optimize the final posterior. This integrated approach enables GeoReward to disentangle variables and achieve precise, market-aware predictions. Detailed Theoretical derivation see Appendix A.

*Table 8.* Synergistic Design Study. Higher is better for both accuracy and sensitivity, and their units are percentage (%).

| Model | MA-RAG | CGVM | SSL | Accuracy | Sensitivity | BR | CL | ES | FR | KR | JP | US | MX | AU | SA |
|---|---|---|---|---|---|---|---|---|---|---|---|---|---|---|---|
| Qwen2-VL-7B (with FC Head) | ✗ | ✗ | ✗ | 55.60 | 36.73 | 53.69 | 56.57 | 50.86 | 54.41 | 53.07 | 59.10 | 56.94 | 53.75 | 58.47 | 59.21 |
| GeoReward (with only MA-RAG) | ✓ | ✗ | ✗ | 56.01 | 37.10 | 53.88 | 56.93 | 52.01 | 54.78 | 53.79 | 59.16 | 57.49 | 53.82 | 58.96 | 59.22 |
| GeoReward (with only CGVM) | ✗ | ✓ | ✗ | 55.92 | 37.18 | 53.71 | 56.27 | 52.38 | 54.89 | 53.39 | 58.67 | 56.88 | 53.56 | 59.10 | 58.64 |
| GeoReward (with only SSL) | ✗ | ✗ | ✓ | 56.04 | 37.12 | 53.76 | 56.06 | 52.48 | 54.47 | 53.36 | 57.78 | 56.32 | 53.68 | 58.65 | 59.48 |
| GeoReward (w/o MA-RAG) | ✗ | ✓ | ✓ | 56.81 | 37.40 | 54.93 | 57.05 | 53.80 | 55.39 | 54.48 | 59.57 | 57.65 | 55.64 | 59.52 | 60.11 |
| GeoReward (w/o CGVM) | ✓ | ✗ | ✓ | 57.98 | 38.95 | 55.50 | 57.85 | 53.78 | 56.75 | 55.19 | 61.47 | 59.39 | 55.98 | 61.71 | 62.21 |
| GeoReward (w/o SSL) | ✓ | ✓ | ✗ | 56.95 | 37.47 | 54.79 | 57.09 | 52.80 | 56.12 | 54.44 | 61.01 | 58.79 | 55.33 | 58.83 | 60.33 |
| **GeoReward** | ✓ | ✓ | ✓ | **60.37** | **40.84** | **58.70** | **61.12** | **56.33** | **59.38** | **57.88** | **63.54** | **61.82** | **58.72** | **63.30** | **62.93** |

# N. MACP Dataset

We evaluate our proposed method on the collected Multi-Country Ad Click Preference (MACP) dataset. We rigorously filtered near-duplicates and identical samples between training and test sets. Specifically, we used product IDs, image perceptual hashes, and textual embeddings to identify and remove any overlapping samples. This prevents the model from leveraging the same-product information across splits. Finally, the dataset comprises 823K training samples and 180K test samples, uniformly distributed across 10 distinct country markets, including "BR", "CL", "ES", "FR", "KR", "JP", "US", "MX", "AU", and "SA". Each sample includes detailed product information (title, category, tags, and other attributes), two ad images for the product (A and B), and the click-through rate (CTR) as a proxy for user preference in the target market. To ensure the confidence level of the click-through rate (CTR), the CTR data is obtained by dividing 30-day cumulative clicks by the corresponding cumulative impressions. The dataset is sourced from a major cross-border e-commerce platform, containing 67K product samples with 250K unique advertising images, ensuring consistency in data source and characteristics.

## N.1. Data Balance

Our MACP dataset demonstrates exceptional statistical balance.

**Sample Balance Across Countries:** We constructed the MACP dataset with exceptional statistical balance across countries (see Appendix for detailed tables). As shown in Table 10, the test set contains exactly $18,055$ samples per country (10.00% each). The training set distribution ranges from 9.34% to 10.57% per country, with minimal variance (0.18%) from perfect equality. This controlled, balanced environment was designed to validate our core hypothesis: that our method can effectively solve the CVE problem when the critical country signal is present in the data.

**Country $\times$ Product Category Distribution:** We provide detailed cross-tabulation tables showing the distribution of product categories across countries for both training (Table 11) and test (Table 12) splits. The training set maintains consistent category representation across all countries. The test set is perfectly balanced, with identical product-category counts across countries, eliminating bias from category–country interactions.

## N.2. Diversity and Generality of the MACP Dataset

The MACP dataset, while sourced from a single major cross-border e-commerce platform, exhibits considerable diversity that supports its use for developing generally applicable models.

**Product Category Diversity:** As shown in Table 11-12, the dataset spans 9 major product categories (e.g., Beauty &

Apparel, Consumer Electronics, Home & Garden, Sports & Entertainment). The distribution of these categories is carefully balanced across all 10 countries (variance $< 0.2\%$ from perfect balance per category), preventing confounding effects between country and product type.

**Geographical and Cultural Coverage:** The 10 countries were strategically selected to cover diverse cultural and economic regions: the Americas (US, BR, MX, CL), Europe (FR, ES), East Asia (JP, KR), the Middle East (SA), and Oceania (AU). No single country dominates the dataset.

**User Base:** While fine-grained user demographics cannot be shared due to privacy policies, the platform serves a global user base numbering in the hundreds of millions. Internal analytics indicate that the user demographics for each country (in terms of age groups, gender distribution, and shopping interests) are representative of national e-commerce trends, suggesting the captured preferences reflect broader consumer behavior.

This combination of product, geographical, and implied user diversity mitigates concerns that our findings are artifacts of a narrow data source and supports the generality of the CVE problem and our proposed solution.

### N.3. Robustness of MA-RAG

Our approach, particularly the MA-RAG component, benefits from the availability of click data knowledge. A legitimate concern is its effectiveness in markets with sparse or outdated data. Fundamentally, this highlights a broader principle: model performance is intrinsically linked to data quality and availability. While sophisticated algorithms can extract patterns from data, they cannot create a signal from its absence. For real-world scenarios with severe geographical imbalance, we employ standard mitigation strategies during the training data preparation for GeoReward. These include targeted oversampling of minority countries or categories and strategic undersampling of majority ones to create a more balanced learning foundation. For emerging markets with very limited initial data, we bootstrap knowledge by leveraging data from culturally or economically similar regions. As more market-specific data accumulates, the MA-RAG component becomes increasingly effective and specialized.

Regarding data "freshness," for the MACP dataset, the training and test sets were collected concurrently, eliminating concerns about data being outdated for the purpose of our experiments. We ensure reliability by calculating the Click-Through Rate (CTR) over a fixed 30-day sliding window prior to the evaluation date. This captures recent user preferences and reduces temporal noise. In a production setting, the knowledge base for MA-RAG is updated periodically (e.g., every 30 days).

### N.4. Data Ethics and Availability

**Data Legality & Privacy Compliance:** The MACP dataset was collected and processed in full compliance with the e-commerce platform's data governance framework and applicable privacy regulations. All data usage has been formally authorized through the platform's research partnership program.

**Anonymization Process:** The dataset has undergone rigorous anonymization: (i) All user identifiers have been removed; (ii) CTR data is aggregated at product-country level; (iii) Personal browsing histories or individual user behaviors are excluded; (iv) Product metadata is limited to publicly available information.

**Geographic Bias Mitigation:** We implemented proactive bias control through stratified sampling across 10 diverse markets and balanced product category representation, as shown in Table 10, as evidenced by our detailed distribution tables (9.34%-10.57% per country in training, exact 10% in testing).

**Data Licensing & Access:** The complete MACP dataset will be publicly released under CC BY-NC 4.0 license upon paper acceptance. The release package includes: (i) Aggregated CTR preferences; (ii) Product images and metadata; (iii) Country-market labels.

**Controlled Access Assurance:** We will release the entire dataset, source code, model configurations, and training scripts to ensure full reproducibility of both GeoReward training and the background generation framework.

### N.5. Country Name Abbreviations

As shown in the Table 9, a comprehensive country abbreviation table is clearly defined, listing all country codes used in our study.

*Table 9.* Country name abbreviations

| Abbreviation | Full Name |
|---|---|
| BR | Brazil |
| CL | Chile |
| ES | Spain |
| FR | France |
| KR | Korea (Republic of Korea) |
| JP | Japan |
| US | United States |
| MX | Mexico |
| AU | Australia |
| SA | Saudi Arabia |

*Table 10.* Per-country sample counts for Train and Test splits

| Country | Train Samples | Train % | Test Samples | Test % |
|---|---|---|---|---|
| Australia (AU) | 76934 | 9.34% | 18055 | 10.00% |
| Brazil (BR) | 86832 | 10.55% | 18055 | 10.00% |
| Chile (CL) | 82074 | 9.97% | 18055 | 10.00% |
| Spain (ES) | 85975 | 10.44% | 18055 | 10.00% |
| France (FR) | 80118 | 9.73% | 18055 | 10.00% |
| Japan (JP) | 78289 | 9.51% | 18055 | 10.00% |
| Korea (KR) | 87066 | 10.57% | 18055 | 10.00% |
| Mexico (MX) | 82351 | 10.00% | 18055 | 10.00% |
| Saudi Arabia (SA) | 82478 | 10.02% | 18055 | 10.00% |
| United States (US) | 81274 | 9.87% | 18055 | 10.00% |
| **Total** | 823391 | 100% | 180550 | 100.00% |

*Table 11.* Country × Product Category distribution for Train split

| Country | AU | BR | CL | ES | FR | JP | KR | MX | SA | US |
|---|---|---|---|---|---|---|---|---|---|---|
| **Watches & Luggage Bags** | 5567 | 6321 | 4547 | 6010 | 5714 | 5809 | 7185 | 5138 | 6814 | 6269 |
| **Beauty & Apparel & Shoes** | 11519 | 13623 | 13320 | 14407 | 13679 | 12107 | 11538 | 15620 | 13049 | 15232 |
| **Consumer Electronics** | 6100 | 7864 | 6613 | 6039 | 5695 | 7099 | 7613 | 6313 | 7043 | 6158 |
| **Home & Garden** | 9365 | 8759 | 11096 | 10796 | 10093 | 8372 | 9281 | 9523 | 8562 | 8837 |
| **Kids & Toys & Hobbies** | 5381 | 5633 | 6358 | 5594 | 5595 | 4817 | 5502 | 5735 | 4289 | 5904 |
| **Sports & Entertainment** | 6329 | 9265 | 7351 | 7960 | 6929 | 8011 | 10094 | 6789 | 5902 | 5316 |
| **Home Appliances & Improvement** | 7903 | 7920 | 7098 | 9328 | 8930 | 6662 | 8222 | 7463 | 9743 | 7521 |
| **Industrial & Technology Tools** | 10632 | 13504 | 10602 | 11016 | 10307 | 10981 | 11989 | 10579 | 12161 | 9713 |
| **Other** | 14138 | 13943 | 15089 | 14825 | 13176 | 14431 | 15642 | 15191 | 14915 | 16324 |
| **Total** | 76934 | 86832 | 82074 | 85975 | 80118 | 78289 | 87066 | 82351 | 82478 | 81274 |

*Table 12.* Country × Product Category distribution for Test split

| Country | AU | BR | CL | ES | FR | JP | KR | MX | SA | US |
|---|---|---|---|---|---|---|---|---|---|---|
| **Watches & Luggage Bags** | 2407 | 2407 | 2407 | 2407 | 2407 | 2407 | 2407 | 2407 | 2407 | 2407 |
| **Beauty & Apparel & Shoes** | 1577 | 1577 | 1577 | 1577 | 1577 | 1577 | 1577 | 1577 | 1577 | 1577 |
| **Consumer Electronics** | 2395 | 2395 | 2395 | 2395 | 2395 | 2395 | 2395 | 2395 | 2395 | 2395 |
| **Home & Garden** | 2375 | 2375 | 2375 | 2375 | 2375 | 2375 | 2375 | 2375 | 2375 | 2375 |
| **Kids & Toys & Hobbies** | 1087 | 1087 | 1087 | 1087 | 1087 | 1087 | 1087 | 1087 | 1087 | 1087 |
| **Sports & Entertainment** | 1059 | 1059 | 1059 | 1059 | 1059 | 1059 | 1059 | 1059 | 1059 | 1059 |
| **Home Appliances & Improvement** | 2504 | 2504 | 2504 | 2504 | 2504 | 2504 | 2504 | 2504 | 2504 | 2504 |
| **Industrial & Technology Tools** | 2896 | 2896 | 2896 | 2896 | 2896 | 2896 | 2896 | 2896 | 2896 | 2896 |
| **Other** | 1755 | 1755 | 1755 | 1755 | 1755 | 1755 | 1755 | 1755 | 1755 | 1755 |
| **Total** | 18055 | 18055 | 18055 | 18055 | 18055 | 18055 | 18055 | 18055 | 18055 | 18055 |

# O. Statistical Significance Testing

We conducted rigorous statistical analysis to validate that the performance improvements of our GeoReward model are not due to random chance.

**Bootstrap Hypothesis Testing:** For each country, we performed $10,000$ bootstrap (Efron, 1979) resamples (with replacement) of the test set. For each bootstrap sample, we calculated the accuracy difference between GeoReward and the strongest baseline (Qwen2-VL-7B with FC Head). The p-value was computed empirically as the proportion of bootstrap samples. As shown in the Figure 7, all improvements reported in Table 1 are statistically significant with $p < 0.001$.

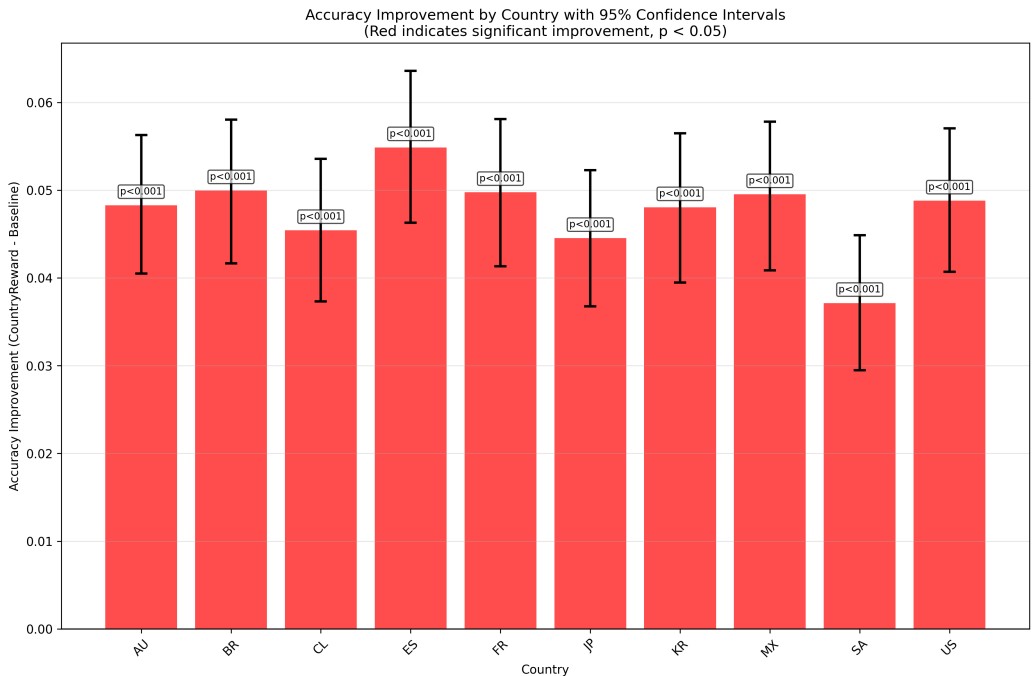

*Figure 7.* Accuracy Improvement by Country with $95\%$ Confidence Intervals.

**Confidence Intervals:** As shown in Table 13, we report $95\%$ confidence intervals (CIs) for all accuracy estimates using the Clopper-Pearson (Clopper & Pearson, 1934) exact method, which is appropriate for binomial proportions. The table below details the per-country performance of the baseline and GeoReward, including their CIs. The results demonstrate that GeoReward's CIs lie entirely above those of the baseline for all countries, confirming a robust and significant improvement.

*Table 13.* Model Accuracy Comparison by Country (with $95\%$ Confidence Intervals)

| Country | Sample Size | Baseline Accuracy (95% CI) | GeoReward Accuracy (95% CI) |
|---------|-------------|----------------------------|------------------------------|
| AU | 18055 | 0.58 (0.58, 0.59) | 0.63 (0.63, 0.64) |
| BR | 18055 | 0.54 (0.53, 0.54) | 0.59 (0.58, 0.59) |
| CL | 18055 | 0.57 (0.56, 0.57) | 0.61 (0.60, 0.62) |
| ES | 18055 | 0.51 (0.50, 0.52) | 0.56 (0.56, 0.57) |
| FR | 18055 | 0.54 (0.54, 0.55) | 0.59 (0.59, 0.60) |
| JP | 18055 | 0.59 (0.58, 0.60) | 0.64 (0.63, 0.64) |
| KR | 18055 | 0.53 (0.52, 0.54) | 0.58 (0.57, 0.59) |
| MX | 18055 | 0.54 (0.53, 0.54) | 0.59 (0.58, 0.59) |
| SA | 18055 | 0.59 (0.58, 0.60) | 0.63 (0.62, 0.64) |
| US | 18055 | 0.57 (0.56, 0.58) | 0.62 (0.61, 0.63) |

