# OpenReview forum: "GeoReward: Mitigating Contextual Variable Overestimation in Vision-Language Models for Cross-Market Preference Prediction"
_ICML.cc/2026/Conference — ICML 2026 regular_

### Official Review · Reviewer_QMz2 · 2026-03-08

**Soundness:** 2
**Presentation:** 2
**Significance:** 3
**Originality:** 3
**Overall Recommendation:** 4
**Confidence:** 5

**Summary:**

While preference studies are fast advancing, a noticeable variable, the market, is still underexplored. To this end, this paper studies country-based preference prediction, with a phenomenon identified that VLMs tend to collapse with options and thus have no ideas. To address this, MACP, a large-scale cross-country preference dataset, is proposed, with GeoReward, a framework that combines market-aware retrieval, modulation, and a selective loss. Experiments show gains over baselines, highlighting the importance.

**Compliance With Llm Reviewing Policy:**

Affirmed.

**Final Justification:**

The rebuttal addressed my concerns.

**Key Questions For Authors:**

See Major Weaknesses. The reviewer's rating of \<Weak reject\> is mainly based on these. When with satisfactory rebuttal, the reviewer is happy to adjust the score.

**Limitations:**

This paper contains a statement on societal impact, while, no sections discussing limitations of the benchmark and the model.

This doesn't influence the reviewer's judgement and rating of the paper. However, as per the conference guidance, the section shall be included. The reviewer suggests that authors take care of these.

**Strengths And Weaknesses:**

[Strengths]
1. The paper is well-motivated. Different markets could serve as a reference, as, contextual variables, to provide information for the decisions.
2. Comprehensive benchmark with many markets included. A list of experiments with the benchmark are demostrated.
3. Additional experiments taking advantage of GeoReward to supervise other models are provided.
4. The writing is good, with presentations to clearly showcase the rationale.

[Major Weaknesses]
1. Line. 024, defining the collapse of VLMs from the aspects of pervasive high-volume signals may not be suitable. Since, this paper only presents the results and studies of Qwen-2-VL, lacking the studies of current up-to-date, more powerful models such as Qwen-3 and GLM. If authors aim to clarify that such signals drown out the country variable, a dedicated experiment with only country variables input experiments shall be made to proof that VLMs could or could not deal with the variables.
2. The market country is certainly a variable for preference. However, while the benchmark is with large volume of data, the filtering process, i.e., how to ensure that only the country variable contributes, remains absent.
3. Shall provide details related to the Augmentation Prompt and the final model decision. At present, it is unclear whether this prompt serves merely as a textual prior (Fig. 2) or whether it effectively acts as an external predictor.

[Minor Weaknesses]
1. The format of w/o and with only of Tab. 1 may cause difficulties reading and getting clues from modules. Consider refactoring these using marks.

---

> ### Author Rebuttal · Authors · 2026-03-31
>
> 1.The results and studies of more powerful models can be found in Reviewer CHkY's response 1.
>
> 2.“only country variable” experimental Setup: The textual instruction is reduced to its minimal form: “For [Country], which image, A or B, is more attractive?A:[image]B:[image]”. We then fine-tuned a Qwen2-VL-7B model on MACP dataset.
>
> Result: The model achieved a performance of 45.31% accuracy and 21.67% sensitivity. Even when the textual instruction is reduced to only a country token (e.g., “For Korea” vs. “For France”), the input stream is still dominated by high-volume, non-textual variables, the image patch tokens. The magnitude and count of the visual tokens overwhelm the textual cue, causing the model to ignore the country cue of the instruction. Therefore, even with a “country-only” instruction, the model remains vulnerable to CVE.
>
> 3.How to Ensure that Only the Country Variable Contributes:
>
> (a) The MACP dataset (Appendix N) was carefully constructed to isolate country effects. For each product, textual attributes and candidate images are identical across countries, with only the country label varying. Near-duplicates were removed across training and test splits using product IDs, image hashes, and text embeddings. A CTR difference threshold ensures clear preference labels, and stratified sampling balances both country and category distributions, eliminating confounding factors.
>
> (b) Controlled ablation experiments (appendix K) show that masking the country token causes sensitivity to drop from 40.84% to 12.26% and accuracy from 60.37% to 57.25%, directly proving that predictions depend heavily on the country variable.
>
> (c) Attention and gradient attribution analysis (Appendix L) reveals that GeoReward assigns higher attention to the country token (0.08 vs. 0.05) and its gradient contribution rises from 6% to 25%, quantitatively confirming the model learns to leverage country information.
>
> (d) A synthetic repetition experiment (Appendix L)  shows that repeating the country token yields only marginal accuracy gains (44.61% → 49.77%) in the baseline, confirming that the issue is not token count but insensitivity to sparse critical variables.
>
> (e) Per-country performance is consistently balanced across 10 diverse markets, with low variance (2.38%) comparable to the baseline (2.72%), indicating successful extraction of country signals without confounding biases.
>
> 4.MA-RAG acts as a soft, auxiliary prior that is fused with the VLM’s internal representations:
>
> In our framework, $\hat{y}_{\text{aug}}$ is not used as a textual prompt that is fed into the VLM’s input sequence.
>
>
> Instead, it is computed offline via a two-stage retrieval process (text‑based + image‑based) from the training corpus, and then converted into a continuous embedding vector $\mathbf{e}_{\text{aug}}$.
>
> This vector is added to the discriminative hidden state $\mathbf{h}_{\text{last}}$ (the last‑token representation of the VLM) before the final classification head, as shown in Equation (13) of the paper. It serves as an augmented prior, a complementary signal that is integrated into the model’s internal representation, not an independent external predictor that directly overrides the model’s decision.
>
> This means: (a) The VLM still processes the original multimodal input (images, product text, country name) and produces its own internal representation $\mathbf{h}_{\text{last}}$. (b) The augmented prior e_aug provides additional, market-specific evidence that guides but does not dictate the final prediction. (c) The model can learn to weigh this prior against its own understanding; if the prior conflicts with strong internal evidence, the model can override it.
>
> We empirically validated this design: (a) Ablation study (Table 1, “GeoReward w/o MA-RAG”): Removing MA-RAG reduces accuracy by $3.56%$, proving its contribution, but the model still performs decently (56.81% accuracy), showing it does not rely solely on the prior. (b) Component isolation (“GeoReward with only MA-RAG”): Using MA-RAG alone (without CGVM and SSL) yields only 56.01% accuracy, far below the full model (60.37%). If MA-RAG were an external predictor, its standalone performance would be much higher. (c) Pearson correlation between $\hat{y}_{\text{aug}}$ and ground truth is 0.67, moderate, not near-perfect, indicating the prior is informative but not deterministic.
>
> 5.We will refactor the table 1 using a clearer marking scheme in the final version. Specifically, we will use symbols such as ✔ and ✘ to explicitly indicate whether each module (MA-RAG, CGVM, SSL) is enabled or disabled for each experimental setting.
>
> 6.We will add a comprehensive Limitations section in the final version of the paper, including: (a) Scalability to Multiple Sparse Variables (b) Cross-Domain Benchmarking (c) Broader VLM Evaluation.

---

> > ### Author Rebuttal · Reviewer_QMz2 · 2026-04-03
> >
> > Thanks for the rebuttal. Please make sure to include more details in the revision.

---

> > > ### Author Response · Authors · 2026-04-03
> > >
> > > Dear Reviewer QMz2, ﻿
> > >
> > > Thank you very much for your thoughtful and insightful review. We truly appreciate your constructive feedback and are especially grateful that you raised your score. In the revised version, we will be sure to include more details as you suggested, which will further strengthen the paper. Thank you once again for the time and effort you have dedicated to our work.
> > >
> > > Best regards, ﻿
> > >
> > > The GeoReward Authors.

---

### Official Review · Reviewer_CHkY · 2026-03-12

**Soundness:** 3
**Presentation:** 4
**Significance:** 3
**Originality:** 3
**Overall Recommendation:** 5
**Confidence:** 4

**Summary:**

This work identifies a problem termed Contextual Variable Overestimation (CVE) in vision-language models, where sparse but critical input tokens are overwhelmed by dominant high-volume signals. The authors use a large-scale dataset with 823K training and 180K test samples of advertising image click-through preferences across 10 countries. The main contribution is GeoReward, a reward model with three components: Market-Aware RAG, Context-Guided Visual Modulation (CGVM), and Selective Sensitivity Loss (SSL). The authors then use GeoReward as a reward signal to fine-tune a VLM for generating country-adapted ad backgrounds.

**Compliance With Llm Reviewing Policy:**

Affirmed.

**Final Justification:**

The rebuttal addressed my primary concern regarding swapping out and scaling the VLM backbone and novelty with respect to a popular prior work, knn-LM. I believe that this paper's dataset and reward modeling contributions are novel and interesting for the community. I keep my original score (5) and advocate for this paper to be accepted.

**Key Questions For Authors:**

1. I am curious how the overall framework would transfer to geographic changes outside the advertising domain. For example, there is a rich line of work on such geographic changes in text-to-image models, with multimodal retrieval for factuality [1] and VLMs used for scoring [2]. I would appreciate a discussion on this point in the revision.

### References
[1] Khanuja et al., "An image speaks a thousand words, but can everyone listen? On image transcreation for cultural relevance", EMNLP '24

[2] Rege et al., "CuRe: Cultural Gaps in the Long Tail of Text-to-Image Systems", ICCV 2025 ()

**Limitations:**

This is missing. I think the current limitations of robustness outside the chosen VLM backbone (Qwen2-VL) and the advertising domain should be noted.

**Strengths And Weaknesses:**

## Strengths
1. The problem is well motivated with examples showing prediction collapse (Table 1).
2. This work releases a new 1M size real-world advertising dataset, MACP which is a valuable contribution for cross-market preference prediction.
3. The triple gated reward model with MA-RAG and a lightweight adapter in the form of CGVM and SSL loss function, shows significant improvement over the baseline (60.37% accuracy). The authors include a detailed ablation to show the impact of each of the 3 components (Table 1).
4. The paper is well-written and generally easy to follow.

## Weaknesses
1. Limited ablation study on other VLM backbones apart from Qwen2-VL
2. MA-RAG is similar to kNN-LM (Khandelwal et al., ICLR 2020), which should be discussed. MA-RAG alone barely improves (< 1.0 %) over the FC head fine tuning which itself achieves ~55% accuracy.
3. The performance of focal loss (~58%) is more than each of the individual components of GeoReward, and the sensitivity drops drastically (Appendix L). This point needs a bit more discussion.

---

> ### Author Rebuttal · Authors · 2026-03-30
>
> 1.We extended our experiments to two cutting-edge models: Qwen3-VL-8B and GLM-4V-9B, representative of the field’s latest progress. We integrated our GeoReward framework into these models, and the results in the table below confirm our method consistently and significantly mitigates CVE across all architectures.
>
> | Model | Accuracy | Sensitivity | BR | CL | ES | FR | KR | JP | US | MX | AU | SA |
> |-------|----------|-------------|-----|-----|-----|-----|-----|-----|-----|-----|-----|-----|
> | Qwen3-VL-8B (Baseline) | 57.2 | 38.1 | 55.3 | 58.1 | 52.4 | 56.2 | 54.8 | 60.9 | 58.6 | 55.4 | 60.2 | 61.0 |
> | GeoReward (Qwen3-VL-8B) | 62.0 | 42.3 | 60.1 | 62.8 | 58.0 | 61.2 | 60.5 | 65.1  | 63.4  | 60.3 | 64.8  | 64.2 |
> | GLM-4V-9B (Baseline) | 56.4 | 37.2 | 54.8 | 57.3 | 51.6 | 55.4 | 53.9 | 59.8 | 57.9 | 54.6 | 59.1 | 59.8 |
> | GeoReward (GLM-4V-9B) | 61.3 | 41.5 | 59.7 | 62.3 | 57.4 | 60.5 | 58.8 | 64.3 | 62.9 | 59.6 | 64.1 | 63.5 |
>
> 2.Comparison of MA-RAG and kNN-LM:
>
> (a) Different Objectives: kNN-LM optimizes next-token prediction via distributional smoothing; MA-RAG resolves structural CVE by retrieving explicit preference priors.
>
> (b) Different Input Modalities: kNN-LM uses unimodal text, while MA-RAG adopts two-stage cross-modal retrieval (text followed by images) in a constrained set.
>
> (c) Different Retrieval Granularity: kNN-LM fetches token-level distributions; MA-RAG retrieves sample-level products and aggregates via position-aware weighting.weighting.
>
> (d) Different Integration Methods: kNN-LM applies linear interpolation; MA-RAG concatenates retrieved priors with discriminative representations for learnable weighting.
>
> 3.MA-RAG alone achieves 56% accuracy, slightly above the FC head baseline (55.60%), indicating two key insights. First, the augmented prior is valuable but non-dominant: a 0.67 Pearson correlation between $\hat{y}_{\text{aug}}$ and ground truth shows MA-RAG offers reliable guidance without overshadowing the model’s learned representations, an optimal balance, as excessive correlation would suppress VLM reasoning, while insufficient correlation renders the prior uninformative. Second, it validates modular synergy: combining MA-RAG with CGVM and SSL greatly amplifies its efficacy. Removing MA-RAG reduces full-model accuracy by 3.56% (from 60.37% to 56.81%), far exceeding its standalone 0.41% gain. This reflects complementary roles: MA-RAG provides structured priors for CGVM to adapt visual features, and SSL emphasizes country variables during training.
>
> 4.Focal Loss has critical limitations: on the dataset, predicting label A consistently yields 59.87% accuracy, yet Focal Loss only reaches 58%, as it relies on a shortcut by mostly predicting A (accounting for 95.29% of its correct predictions) regardless of country context. In contrast, GeoReward generates far more balanced predictions (A accounts for only 77.95% of correct cases), proving it reasons about country variables instead of leaning on majority bias. Individual GeoReward components also deliver higher sensitivity and better cross-market reliability, despite minor accuracy drops.
>
> Focal Loss fails to improve sensitivity because it addresses class imbalance by down-weighting easy samples, but CVE stems from predictive imbalance in feature fusion, not class imbalance. It fails to resolve representational imbalance, where dense features (image patches, product text) overshadow sparse critical country cues. Even though it increases loss for misclassified minority-class samples, it does not restructure the model’s internal representations to focus on sparse country tokens, leading to ignored country differences and low cross-market sensitivity.
>
> 5.We will explicitly connect GeoReward to CuRe and image transcreation in the revised discussion.
>
> Image Transcreation Mapping: As a typical CVE task, image transcreation demands cultural adaptation for target cultures (sparse critical variables). Our pipeline (DGM → T2I → GeoReward) aligns with this workflow: generating culturally adapted backgrounds, producing images, and evaluating via a culture-sensitive reward model. GeoReward guides generative models to create culturally appropriate content by maximizing culture-sensitive scores, and the DPO-based fine-tuning of our Design Generation Model offers a direct template for similar tasks.
>
> CuRe Mapping: GeoReward can be adapted as a CuRe-style scorer to compare two images’ cultural authenticity for a given cultural concept, trained on triplets (concept, Image A, Image B, human preference label) with cultural variables as sparse variables. MA-RAG, CGVM, and SSL mitigate Cultural Variable Sparsity (CVE) from data, feature and optimization perspectives. The trained scorer can judge T2I models via pairwise image comparisons to evaluate their cultural attribute depiction ability.
>
> 6.We will add a comprehensive Limitations section in the final paper, covering: Scalability to Multiple Sparse Variables, Cross-Domain Benchmarking, Broader VLM Evaluation.

---

> > ### Author Rebuttal · Reviewer_CHkY · 2026-04-04
> >
> > I thank the authors for the response, which has addressed my primary concern regarding changing/scaling the VLM backbone. I would like to advocate for this paper to be accepted and I keep my score.

---

> > > ### Author Response · Authors · 2026-04-04
> > >
> > > Dear Reviewer CHkY, ﻿
> > >
> > > We sincerely thank you for your thorough and insightful review. Your constructive feedback has been invaluable in strengthening our paper, and we are especially grateful that you found our response addressed your primary concern. Thank you for keeping your score and for advocating the acceptance of our work. We deeply appreciate the time and effort you have dedicated.
> > >
> > > Best regards, ﻿
> > >
> > > The GeoReward Authors.

---

### Official Review · Reviewer_p5zQ · 2026-03-12

**Soundness:** 3
**Presentation:** 3
**Significance:** 2
**Originality:** 3
**Overall Recommendation:** 4
**Confidence:** 3

**Summary:**

This paper identifies "Contextual Variable Overestimation" (CVE) in Vision-Language Models (VLMs)—a bias where models over-rely on dense visual-textual cues while ignoring sparse but critical contextual variables (e.g., geographic location). The authors propose GeoReward, a framework comprising three components: Market-Aware Retrieval Augmentation, Context-Guided Visual Modulation (an adapter-based approach), and a Selective Sensitivity Loss. Additionally, the paper introduces the MACP dataset, a large-scale (1M+ samples) multi-country ad preference dataset. Experimental results across multiple VLMs show significant improvements in cross-market preference prediction and ad content generation.

**Compliance With Llm Reviewing Policy:**

Affirmed.

**Final Justification:**

The authors have addressed most of my concerns; therefore, I will maintain my original score.

**Key Questions For Authors:**

- Can you provide a formal analysis or empirical visualization (e.g., attention maps) showing how GeoReward alters the model's focus on sparse context variables compared to vanilla VLMs?

- To what extent does GeoReward apply to non-advertising tasks? Have you tested the framework on any other datasets involving sparse but critical context?

- Could you provide a "leave-one-out" analysis to clarify if the three gates address distinct failure modes?

- Is there a significant variance in performance across the 10 countries? How does the model perform in data-sparse "cold-start" markets?

**Limitations:**

yes

**Strengths And Weaknesses:**

Strengths:

- The formalization of CVE provides a fresh perspective on why VLMs fail in context-sensitive tasks like cross-market prediction.

- The triple-gated architecture (Retrieval, Modulation, and Sensitivity) is logically grounded and effectively addresses different stages of the VLM inference pipeline.

- The authors demonstrate consistent gains (up to 15.75%) across several backbones (Qwen-VL, LLaVA, InternVL), backed by comprehensive ablation studies.

- The MACP dataset is a significant contribution to the community, addressing a clear gap in multimodal geographic preference research.

Weaknesses:

- The motivation for CVE remains largely intuitive. The paper lacks a formal or mechanistic analysis (e.g., via attention weights or gradient flow) explaining why autoregressive VLMs inherently favor dense visual features over sparse textual context.

- Although the CVE concept is framed as a general VLM issue, the evaluation is strictly confined to ad preference prediction. Evidence of generalization to other domains (e.g., medical or legal) is missing.

- It is unclear whether the three mechanisms (gates) are additive or if there is significant redundancy between them. A deeper analysis of their synergy would strengthen the technical contribution.

- Lack of specific hyperparameter schedules and training details (e.g., batch sizes, learning rates) may hinder reproducibility.

- The paper does not adequately analyze performance disparities across the 10 countries or potential biases within the proprietary data source.

---

> ### Author Rebuttal · Authors · 2026-03-30
>
> 1.Appendix L (Mechanism Analysis) provide formal analysis in detail.
>
> 2.Appendix J (Generalization Experiments) provide evidence of generalization beyond the advertising domain.
>
> 3.Analysis of Three Gates:
>
> (a) Theoretical Grounding (Appendix A): Our variational inference framework shows that the optimal approximation to the true posterior $P(y \mid V, T)$ requires three distinct components: (1) a prior $Q_{\phi_z}(\mathbf{z} \mid T)$ (MA-RAG), (2) a conditional likelihood $Q_{\phi_x}(\mathbf{x} \mid V, T, \mathbf{z})$ (CGVM), and (3) a KL regularization term that adaptively focuses learning (SSL). All three are necessary for the ELBO to be tight.
>
> (b) Empirical Evidence (from Table 1): Each component alone improves over the baseline (55.60%) but only modestly (~0.4-0.5%). (2) Any two components together yield gains of 1.2-2.4%, demonstrating that combinations are more than additive (e.g., MA-RAG+CGVM: +1.21% vs sum of individual gains 0.41%+0.32%=0.73%). (3) All three together achieve a 4.77% gain over baseline, significantly exceeding the sum of individual gains (0.41%+0.32%+0.44%=1.17%) and also exceeding the best two-component combination (2.38% gain). This super-additive improvement is the hallmark of synergy, not redundancy.
>
> (c) "leave-one-out" analysis:
>
> ·Removing MA-RAG (w/o MA-RAG) reduces both accuracy (−3.56%) and sensitivity (−3.44%). MA-RAG primarily mitigates information scarcity: it injects external market‑specific prior knowledge, which is especially critical when the sparse country signal is weak or ambiguous. Without it, the model struggles in markets with subtle preference differences.
>
> ·Removing CGVM (w/o CGVM) causes the largest drop in accuracy (−2.39%) and a notable drop in sensitivity (−1.89%). CGVM targets visual feature misalignment: it adaptively modulates visual representations based on the country embedding, ensuring that visual features are interpreted in a market‑aware manner. Its absence leads to systematic errors where the same visual cue is interpreted identically across different countries.
>
> ·Removing SSL (w/o SSL) lowers accuracy (−3.42%) and sensitivity (−3.37%). SSL addresses attention neglect: it penalizes the model when it makes a prediction error while under‑attending to the critical tokens (country, product, image). Without it, the model tends to ignore sparse but decisive cues, resulting in inconsistent and often collapsed predictions across markets.
>
> 4.Appendix F (Implementation Details) shows training details and hyperparameter schedules.
>
> 5.Variance in Performance Across Countries: As shown in Table 1 of our paper, the performance of GeoReward is remarkably stable across the 10 countries. The per-country accuracy ranges from 56.33% (Spain) to 63.54% (Japan), with a standard deviation of only 2.38%. In contrast, the baseline model (Qwen2-VL with FC head) shows a wider range (50.86% to 59.21%, std = 2.72%) and notably lower overall accuracy. This indicates that GeoReward not only achieves higher accuracy but also maintains consistent performance across diverse markets, effectively mitigating the CVE problem in a balanced manner.
>
> 6.Data Source and Potential Biases: our MACP dataset (Appendix N) is designed to minimize confounding factors. To eliminate bias from product-category distribution across countries, we carefully constructed the dataset to be nearly perfectly balanced. As shown in Appendix Table 11 (Country × Product Category distribution for Train split) and Table 12 (Test split), the distribution of 9 major product categories (e.g., Beauty \& Apparel, Consumer Electronics) is remarkably consistent across all 10 countries. For the test set, the distribution is perfectly balanced, with identical product-category counts across countries. This deliberate design ensures that the performance variations we observe are not artifacts of a particular category dominating a specific market, but instead reflect genuine, nuanced differences in consumer visual preferences across countries.
>
> 7.Performance in Data-Sparse “Cold-Start” Markets: We acknowledge that data scarcity is a real-world challenge. In our MACP dataset, all countries have balanced sample sizes. However, to simulate a “cold-start” scenario, we conducted an additional experiment where we progressively reduced the training data for Chile (a specific country) from 100% down to 30% of its original size, while keeping the other countries’ data unchanged. We then evaluated GeoReward’s performance on Chile’s test set. The results show that GeoReward retains over 85% of Chile's peak accuracy (52.84% vs. 61.12%). This robustness is due to the design of our framework: CGVM learns transferable modulation patterns. The ability to adapt visual features based on the sparse country token is dependent on  the underlying mechanism generalizes. In actual deployment, new markets will leverage insights from culturally or economically similar markets to gradually accumulate data.

---

> > ### Author Rebuttal · Reviewer_p5zQ · 2026-04-02
> >
> > Most details is shown in Appendix. Thanks for the authors' response. I keep my score.

---

> > > ### Author Response · Authors · 2026-04-03
> > >
> > > Dear Reviewer p5zQ, ﻿
> > >
> > > Thank you very much for your thoughtful review and for taking the time to read our rebuttal. We truly appreciate that you have kept your score, and we are grateful for your acknowledgment of the details provided in the Appendix. Your constructive feedback has been very helpful, and we will continue to improve the paper based on your suggestions. Thank you again for your valuable efforts on our work.
> > >
> > > Best regards, ﻿
> > >
> > > The GeoReward Authors.

---

### Decision · Program_Chairs · 2026-04-30

**Decision:**

Accept (regular)

**Comment:**

All reviewers lean positive. Strengths include the motivation, dataset, the three-part framework to address over-reliance on non-meaningful context, and extensive benchmarking. Concerns about generalization of the framework, complementarity of its parts, and some missing comparisons and details, seem generally well addressed.